



**Improving thermodynamic profile retrievals from microwave**
**radiometers by including Radio Acoustic Sounding System (RASS)**
**observations**
Irina V. Djalalova[1,2], David D. Turner[3], Laura Bianco[1,2],
James M. Wilczak[2], James Duncan[1,2], Bianca Adler[1,2] and Daniel Gottas[2]
[1] Cooperative Institute for Research in Environmental Sciences (CIRES), Boulder, CO, USA
[2] National Oceanic and Atmospheric Administration, Physical Sciences Laboratory, Boulder, CO, USA
[3] National Oceanic and Atmospheric Administration, Global Systems Laboratory, Boulder, CO USA

18 ---------------------------

19 Corresponding author address: Irina V. Djalalova (Irina.V.Djalalova@noaa.gov), NOAA/Physical

20 Science Laboratory, 325 Broadway, mail stop: PSD3, Boulder, CO 80305. Tel.: 303-497-6238.

21 Fax: 303-497-6181.





**Abstract**
Thermodynamic profiles are often retrieved from the multi-wavelength brightness
temperature observations made by microwave radiometers (MWRs) using regression methods
(linear, quadratic approaches), artificial intelligence (neural networks), or physical-iterative
methods. Regression and neural network methods are tuned to mean conditions derived from
a climatological dataset of thermodynamic profiles collected nearby. In contrast, physical-
iterative retrievals use a radiative transfer model starting from a climatologically reasonable
value of temperature and water vapor, with the model run iteratively until the derived
brightness temperatures match those observed by the MWR within a specified uncertainty.
In this study, a physical-iterative approach is used to retrieve temperature and humidity
profiles from data collected during XPIA (eXperimental Planetary boundary layer Instrument
Assessment), a field campaign held from March to May 2015 at NOAA's Boulder Atmospheric
Observatory (BAO) facility. During the campaign, several passive and active remote sensing
instruments as well as in-situ platforms were deployed and evaluated to determine their
suitability for the verification and validation of meteorological processes. Among the deployed
remote sensing instruments was a multi-channel MWR, as well as two radio acoustic sounding
systems (RASS), associated with 915-MHz and 449-MHz wind profiling radars.
Having the possibility to combine the information provided by the MWR and RASS
systems, in this study the physical-iterative approach is tested with different observational
inputs: first using data from surface sensors and the MWR in different configurations, and then
including data from the RASS. These temperature retrievals are also compared to those derived
by a neural network method, assessing their relative accuracy against 58 co-located radiosonde



profiles. Results show that the combination of the MWR and RASS observations in the physical-
iterative approach allows for a more accurate characterization of low-level temperature
inversions, and that these retrieved temperature profiles match the radiosonde observations
better than all other approaches, including the neural network, in the atmospheric layer
between the surface and 5 km AGL.  Specifically, in this layer of the atmosphere, both root
mean square errors and standard deviations of the difference between radiosonde and
retrievals that combine MWR and RASS are improved by ~0.5 $^{o}$C compared to the other
methods. Pearson correlation coefficients are also improved.
















## 1. Introduction


To monitor the state of the atmosphere for process understanding and for model
verification and validation, scientists rely on observations from a variety of instruments, each
one having its set of advantages and disadvantages. Using several diverse instruments allows
one to monitor different aspects of the atmosphere, while combining them in an optimized
synergetic approach can improve the accuracy of the information we have on the state of the
atmosphere.
During the eXperimental Planetary boundary layer Instrumentation Assessment (XPIA)
campaign, an U.S. Department of Energy sponsored experiment held at the Boulder
Atmospheric Observatory (BAO) in Spring 2015, several instruments were deployed (Lundquist
et al., 2017) with the goal of assessing their capability for measuring flow within the
atmospheric boundary layer. XPIA investigated novel measurement approaches, and quantified
uncertainties associated with these measurement methods. While the main interest of the XPIA
campaign was on wind and turbulence, measurements of other important atmospheric
variables were also collected, including temperature and humidity. Among the deployed
instruments were two identical microwave radiometers (MWRs) and two radio acoustic
sounding systems (RASS), as well as radiosondes launches that were used for verification.
MWRs are passive sensors, sensitive to atmospheric temperature and humidity content
that allow for a high temporal observation of the state of the atmosphere, with some
advantages and limitations. In order to estimate profiles of temperature and humidity, they
observe atmospheric brightness temperature and apply radiative transfer equations
(Rosenkranz, 1998) and neural network retrievals (Solheimet al., 1998a, and 1998b; Ware et al.,
2003), or physical retrieval methodologies that can include more information about the
atmospheric state in the retrieval process (Turner and Blumberg, 2019). Advantages of MWRs
include their compact design, the relatively high temporal resolution of the measurements (2-3
minutes), the possibility to observe the vertical structure of both temperature and moisture,
the deep layer of the atmosphere that can be monitored including during cloudy conditions,
and their capability to operate in a standalone mode. Disadvantages include the limited
accuracy, as the temperature and humidity profiles are not actively measured but retrieved,
their lower accuracy in the presence of rain because of scattering of radiation due to raindrops
in the atmosphere (and because some water can still deposit on the radome, although the
instruments use a hydrophobic radome and force airflow over the surface of the radome during
rain), rather coarse vertical resolution, and for retrievals the necessity to have a site specific
climatology. Other disadvantages include the challenges related to performing accurate
calibrations (Küchler et al., 2016, and references within), radio frequency interference (RFI), and
the low accuracy on the retrieved liquid water path (LWP) especially for values of LWP less than
50 g/m$^2$.

RASS, in comparison, are active instruments that emit a longitudinal acoustic wave

upward, causing a local compression and rarefaction of the ambient air. These density
variations are tracked by the Doppler radar associated with the RASS, and the speed of the
propagating sound wave is measured. The speed of sound is related to the virtual temperature
Tv (North et al., 1973), and therefore, RASS are routinely used to remotely measure vertical
profiles of virtual temperature in the boundary layer. Being an active instrument, the RASS is in
general more accurate than a passive instrument (Bianco et al., 2017), but they also come with





their sets of disadvantages. The main limitations of RASS for retrieval purposes are its low
temporal resolution (typically a 5-min averaged RASS profile is measured once or twice per
hour), and their limited altitude coverage. Recent studies (Adachi and Hashiguchi, 2019) have
shown that to make them more suitable to operate in urban areas RASS could use parametric
speakers to take advantage of their high directivity and very low side lobes. Nevertheless, the
maximum height reached by the RASS is still limited, being a function of both radar frequency
and atmospheric conditions (May and Wilczak, 1993), and is determined both by the
attenuation of the sound, which is a function of atmospheric temperature, humidity, and
frequency of the sound source, and the advection of the propagating sound wave out of the
radar's field-of-view. Therefore, data availability is usually limited to the lowest several km,
dependent on the frequency of the radar. In addition, wintertime coverage is usually
considerably lower than that in summer, due to a higher probability of stronger winds
advecting the sound wave away from the radar in the winter.

To get a better picture of the state of the temperature and moisture structure of the

atmosphere, it makes sense to try to combine the information obtained by both MWR and
RASS. Integration of different instruments has been of scientific interest for several years (Han
and Westwater 1995; Stankov et al. 1996; Bianco et al., 2005; Engelbart et al., 2009; Cimini et
al., 2020, Turner and Löhnert, 2020, to name some). In this study we particularly focus on the
combination of the MWR and RASS observations in the retrievals to improve the accuracy of
the temperature profiles in the lowest 5 km compared to the standard MWR retrievals
obtained through neural network (NN) processing, or compared to physical retrieval
approaches that do not include the information from RASS measurements.



This paper is organized as follows: Section 2 summarizes the experimental dataset;
Section 3 introduces the principles of the physical retrieval approaches used to obtain vertical
profiles of the desired variables; Section 4 produces statistical analysis of the comparison
between the different retrieval approaches and radiosonde measurement; finally, conclusions
are presented in Section 5.

**2. XPIA data**
The data used in our analysis were collected during the XPIA experiment, held in Spring
2015 (March-May) at the NOAA's Boulder Atmospheric Observatory (BAO) site, in Erie,
Colorado (Lat.: 40.0451 N, Lon.: 105.0057 W, El.: 1584 m MSL). XPIA was the last experiment
conducted at this facility, as after almost 40 years of operations the BAO 300-m tower was
demolished at the end of 2016 (Wolfe and Lataitis, 2018). XPIA was designed to assess the
capability of different remote sensing instruments for quantifying boundary layer structure, and
was a preliminary study as many of these same instruments were later deployed, among other
campaigns, for the second Wind Forecast Improvement Project WFIP2 (Shaw et al., 2019;
Wilczak et al., 2019) which investigated flows in complex terrain for wind energy applications,
and were for example used to study cold air pool and gap flow characteristics (Adler et al.,
2020; Banta et al., 2020; Neiman et al., 2019). The list of the deployed instruments included
active and passive remote-sensing devices, and in-situ instruments mounted on the BAO tower.
Data collected during XPIA are publicly available at https://a2e.energy.gov/projects/xpia. A
detailed description of the XPIA experiment can be found in Lundquist et al. (2017), while a
specific look at the accuracy of the instruments used in this study can be found in Bianco et al.

(2017).


**2.1 MWR measurements**

Two identical MWRs, managed by NOAA (MWR-NOAA) and by the University of
Colorado (MWR-CU), were deployed next to each other at the visitor center ~600 m south of
the BAO tower (see Lundquist et al., 2017 for a detailed map of the study area). Both MWRs
have 35-channels spanning a range of frequencies, with 21 channels in the lower (22-30 GHz) K-
band frequency band, and 14 channels in the higher (51-59 GHz) V-band frequency band.
Frequencies in the K-band are more sensitive to water vapor and cloud liquid water, while
frequencies in the V-band are sensitive to atmospheric temperature due to the absorption of
atmospheric oxygen (Cadeddu et al., 2013).  Both MWRs observed at the zenith and at 15- and
165-degree elevation angles in the north-south plane (referred to as oblique elevation scans
hereafter; note zenith views have 90-degree elevation angles). In addition, each MWR was
provided with a separate surface sensor to measure pressure, temperature, and relative
humidity at the installation level that was ~2.5 m above ground level (AGL). MWRs are passive
devices which record the natural microwave emission in the water vapor and oxygen
absorption bands from the atmosphere, providing measurements of the brightness
temperatures. Vertical profiles of temperature (T), water vapor density (WVD), and relative
humidity (RH) were retrieved in real-time during XPIA every 2-3 minutes using a NN approach
provided by the private manufacturing company Radiometrics (Solheim et al. 1998). The NN
used a training dataset based on a 5-year climatology of profiles from radiosondes launched at



the Denver International Airport, 35 miles south-east from the XPIA site. NN-based MWR
vertical retrieval profiles were obtained using the zenith and an average of two oblique
elevation scans, all extending for 58 levels up to 10 km, with nominal vertical levels depending
on the height (every 50 m from the surface to 500 m, every 100 m from 500 m to 2 km, and
every 250 m from 2 to 10 km, AGL). In this study we make use of the NN zenith and of the NN
oblique, where the latter can average out small-scale horizontal inhomogeneities of the
atmosphere.

The MWR-CU operated from 9 March to 7 May 2015, while MWR-NOAA was unavailable

between 5-27 April 2015. For the overlapping dates, temperature retrieved from the two
MWRs showed very good agreement with less than 0.5 K bias and 0.994 correlation (Bianco et
al., 2017). For this reason, we use only the MWR-CU (hereafter simply called MWR).

**2.2 Radiosonde measurements**

Between 9 March and 7 May 2015, while the MWR was operational, radiosondes were

launched by the National Center for Atmospheric Research (NCAR) assisted by several students
from the University of Colorado over three selected periods, one each in March, April, and May.
There was a total of 59 launches, mostly four times per day, around 14:00, 18:00, 22:00 and
0200 UTC (8:00, 12:00, 16:00 and 20:00 local standard time, LST). All radiosondes were Vaisala
RS92. The first 35 launches, between 9-19 March, were done from the visitor center, while the
11 launches, between 15-22 April, and 13 launches, between 1-4 May, were done from the
water tank site, ~1000 meters apart (see Lundquist et al., 2017 for a detailed map of the study
area). The radiosonde measurements included temperature, dewpoint temperature, and



220 relative humidity, to altitudes usually higher than 10 km AGL, with measurements every few

221 seconds.


223 **2.3 WPR-RASS measurements**

224  Two NOAA wind profiling radars (WPRs), operating at frequencies of 915-MHz and 449-

225 MHz, were deployed at the visitor center (same location of the MWR) during XPIA. These

226 systems are primarily designed to measure the vertical profile of the horizontal wind vector, but

227 co-located RASS also observe profiles of virtual temperature in the lower atmosphere, with

228 different resolutions and height coverages depending on the WPR. Thus, the RASS associated

229 with the 915-MHz WPR (hereafter referred to as RASS 915) measured virtual temperature from

230 120 to 1618 m with a vertical resolution of 62 m, and the 449 MHz RASS (hereafter referred to

231 as RASS 449) sampled the boundary layer from 217 to 2001 m with a vertical resolution of 105

232 m. The maximum height reached by the RASS is a function of both radar frequency and

233 atmospheric conditions (May and Wilczak, 1993), and is usually lower for RASS 915 data, as will

234 be shown later in the analysis.

235  The RASS data were processed using a radio frequency interference (RFI)-removal

236 algorithm (performed on the RASS spectra), a consensus algorithm (Strauch et al. 1984)

237 performed on the moment data using a 60% consensus threshold, a Weber-Wuertz outlier

238 removal algorithm (Weber et al., 1993) performed on the consensus averages, and a RASS

239 range-correction algorithm (Görsdorf and Lehmann, 2000) using an average relative humidity

240 setting of 50% determined from the available observations.






**2.4 BAO data**
The BAO 300-m tower was built in 1977 to study the planetary boundary layer (Kaimal
and Gaynor 1983). During XPIA, measurements were collected at the surface (2 m) and at six
higher levels (50, 100, 150, 200, 250 and 300 m AGL).  Each tower level was equipped with 2
sonic anemometers on orthogonal booms, and one sensor based on a Sensiron SHT75 solid-
state sensor to measure temperature and relative humidity with a time resolution of 1 s, and
averaged over five minutes.
The observational temperature and water vapor surface data are used from the more
accurate observations at the BAO tower 2 m AGL level (Horst, 2016), to replace the data
measured by the less accurate MWR inline surface sensor.


**3.   Physical retrievals**
Other than NN approaches, a physical retrieval (PR) iterative approach can be used to
retrieve vertical profiles of thermodynamic properties from the MWR observations (Maahn et
al 2020). In this case, using a radiative transfer model, the process starts with a climatologically
reasonable value of temperature and water vapor, and is iteratively repeated until the
computed brightness temperatures match those observed by the MWR within the uncertainty
of the observed brightness temperatures (Rodgers, 2000; Turner and Löhnert, 2014; Maahn et
al. 2020).



### 3.1 Iterative retrieval technique


For this study, the physical retrieval (PR) uses a microwave radiative transfer model,
MonoRTM (Clough et al., 2005), which is fully functional for the microwave region and was
intensively evaluated previously on MWR measurements (Payne et al. 2008; 2011). We start
with the state vector $X_a$ = [**T, Q,** LWP]$^T$, where superscript T denotes transpose. **T** (K) and **Q**
(g/kg) are temperature and water vapor mixing ratio profiles at 55 vertical levels from the
surface up to 17 km, with the distance between the levels increasing exponentially-like with
height. LWP is the liquid water path in (g/m$^2$) that measures the integrated content of water in
the entire vertical column above the MWR, and is a scalar. For this study we have $X_a$ with
dimensions equal to 111 x 1 (two vectors **T** and **Q** with 55 levels each, and LWP).  We are using
the retrieval framework of Turner and Blumberg (2019), but only using MWR data (no spectral
infrared) and will augment the retrieval to include RASS profiles of Tv.
The observation vector **Y** from the beginning includes temperature and water vapor
mixing ratio measured at the surface, and brightness temperature (**Tb**) measured by the MWR.
The MonoRTM model **F** is used as the forward model to estimate the observation vector **Y** from
the current state vector **X**, from Eq. (1), iterating until the difference between **F(X)** and **Y** is
small within a specified uncertainty:
$$X_{n+1} = X_a + (S_a^{-1} + K^T S_\varepsilon^{-1} K)^{-1} K^T S_\varepsilon^{-1} [Y - F(X_n) + K(X_n - X_a)] \quad (1)$$
with:
$$X_a = \begin{bmatrix} T \\ Q \\ L \end{bmatrix} \qquad S_a = \begin{bmatrix} \sigma_{TT}^2 & \sigma_{TQ}^2 & 0 \\ \sigma_{QT}^2 & \sigma_{QQ}^2 & 0 \\ 0 & 0 & \sigma_L^2 \end{bmatrix} \qquad K_{ij} = \frac{\partial F_i}{\partial X_j}$$

$$S_\varepsilon = \begin{bmatrix} \sigma^2_{T\,sfc} & 0 & 0 & 0 & 0 \\ 0 & \sigma^2_{Q\,sfc} & 0 & 0 & 0 \\ 0 & 0 & \sigma^2_{Tb_{zenith}} & 0 & 0 \\ 0 & 0 & 0 & \sigma^2_{Tb_{zenith+oblique}} & 0 \\ 0 & 0 & 0 & 0 & \sigma^2_{Tv_{RASS915(449)}} \end{bmatrix}$$




and **Y**, depending on the configuration used, being equal to:

$$Y_1 = \begin{bmatrix} T_{sfc} \\ Q_{sfc} \\ Tb_{zenith} \end{bmatrix} \qquad Y_2 = \begin{bmatrix} T_{sfc} \\ Q_{sfc} \\ Tb_{zenith} \\ Tb_{zenith+oblique} \end{bmatrix}$$


$$Y_3 = \begin{bmatrix} T_{sfc} \\ Q_{sfc} \\ Tb_{zenith} \\ Tb_{zenith+oblique} \\ Tv_{RASS915} \end{bmatrix} \qquad Y_4 = \begin{bmatrix} T_{sfc} \\ Q_{sfc} \\ Tb_{zenith} \\ Tb_{zenith+oblique} \\ Tv_{RASS449} \end{bmatrix}$$


The superscripts T and -1 indicate transpose or inverse matrix, respectively. Also,
vectors and matrices are shown in bold. Note that we are including the 2-m surface-level
observations of temperature and water vapor mixing ratio (Tsfc and Qsfc, respectively) as part
of the observation vector Y, and thus the uncertainties in these observations are included in $S_\varepsilon$.
The first guess of the state vector **X**, **X₁** in Eq. (1), is set to be equal to the mean state
vector of climatological estimates, or a "prior" vector **Xₐ**, which is calculated independently for
each month of the year from climatological sounding profiles (10 years) in the Denver area.
**Sₐ** is the covariance matrix of the "prior" vector that includes not only temperature or water
vapor variances but also the covariances between them. **K** is the Jacobian matrix, computed
using finite differences by perturbing the elements of X and rerunning the radiative transfer
model.



We start with four configurations for the observational vector $\mathbf{Y}$ ($\mathbf{Y_1}$, $\mathbf{Y_2}$, $\mathbf{Y_3}$, and $\mathbf{Y_4}$). The

MWR provides the $\mathbf{Tb}$ measurements in all schemes, zenith only in $\mathbf{Y_1}$ (which also includes the
2-m in-situ observations of temperature and humidity), and zenith and oblique in $\mathbf{Y_2}$, $\mathbf{Y_3}$, and $\mathbf{Y_4}$.
Using additional measurements from the co-located radar systems with RASS, we may further
expand the observational vector with either RASS 915 ($\mathbf{Y_3}$) or RASS 449 ($\mathbf{Y_4}$) virtual temperature
observations. The covariance matrix of the observed data, $\mathbf{S_\varepsilon}$, depends on the chosen $\mathbf{Y_i}$ as it is
highlighted by the red numbers in the matrix description, with increasing dimensions from $\mathbf{Y_1}$ to
$\mathbf{Y_2}$ and additional increasing dimensions to $\mathbf{Y_3}$ and $\mathbf{Y_4}$ through the multi-level measurements of
the RASS (Turner and Blumberg, 2019). Table 1 summarizes the observational information
included in these four different configurations of the PR.

|  | $T_{sfc}$ | $Q_{sfc}$ | $Tb_{zenith}$ | $Tb_{zenith-oblique}$ | $Tv_{RASS915}$ | $Tv_{RASS449}$ |
|---|---|---|---|---|---|---|
| $\mathbf{Y_1}$ = MWRz | X | X | X |  |  |  |
| $\mathbf{Y_2}$ = MWRzo | X | X | X | X |  |  |
| $\mathbf{Y_3}$ = MWRzo915 | X | X | X | X | X |  |
| $\mathbf{Y_4}$ = MWRzo449 | X | X | X | X |  | X |

*Table 1. Four PR configurations corresponding to the four observational $Y_i$ vectors in Eq. (1).*

We assume that there is no covariance between different instruments as well as

between different channels (MWR) or height levels (RASS) of each instrument, therefore this
covariance matrix $\mathbf{S_\varepsilon}$ is diagonal. The Jacobian matrix, $\mathbf{K}$, has dimensions m x 111, where m is
the length of the vector $\mathbf{Y_i}$, therefore its dimensions increase correspondingly with the inclusion
of more observational data. $\mathbf{K}$ makes the "connection" between the state vector and the



observational data and should be calculated at every iteration.

**3.2 Bias-correction**
Observational errors propagate through the retrieval into the derived profiles (i.e. the
bias of the observed data will contribute to bias the retrievals.) For that, retrieval uncertainties
in Eq. (1) from $Y = Y_1$ or $Y_2$ derive only from uncertainties in surface and MWR data, while
retrieval uncertainties from $Y = Y_3$ or $Y_4$ are coming from uncertainties in surface, MWR, and
RASS measurements.
While the bias of the retrieval depends on both the sensitivity of the forward model and
the observational uncertainty, we can try to eliminate, or at least to reduce, the systematic
error in the MWR observations. To this aim, we first looked for clear sky days (to reduce the
degrees of freedom associated with clouds) during the period of the measurements. One
method to identify clear-sky times is to use brightness temperature observations in the 30 GHz
water vapor sensitive channel. The random uncertainty in brightness temperature was
calculated as its standard deviation during clear sky times and for this channel is approximately
0.3 K (but during periods with liquid-bearing clouds overhead, the standard deviation of the 30
GHz Tb is markedly higher than this threshold due to the non-homogeneous nature of clouds
and thus their contribution to the downwelling microwave radiance). Four clear-sky days were
selected, March 10 and 30, and April 13 and 29. The bias was then computed on all channels
over these selected clear-sky days and removed from all measurements. Fig. 1 shows the
results of the bias-correction for the four chosen clear-sky days. The green lines on this figure
indicate the MWR random errors at each frequency calculated as the standard deviation of Tb



averaged over one-hour sliding window; these are 0.3-0.4 K for K-band channels and 0.6-0.7 K
for V-band channels.

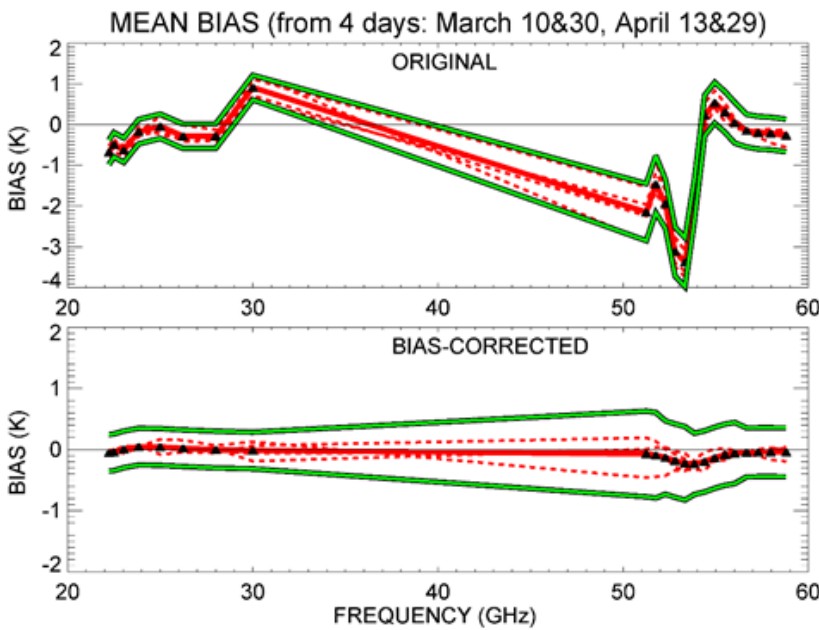


*Fig.1. Bias for the four chosen clear-sky days (red-dashed lines) and their mean (red solid line)*
*for the original observations in the top panel, and for the bias-corrected data in the bottom*
*panel. Green lines are the uncertainty boundaries around the mean bias. Frequencies used in the*
*PR algorithm are marked with black triangles in both panels.*

This bias correction was applied to the brightness temperature used in the PR approach;
however, the NN retrievals used the uncorrected brightness temperature, since it was non-
trivial for us to reprocess those retrievals.

The retrieved profiles of the four different PR configurations presented in Table 1

(MWRz, MWRzo, MWRzo915, MWRzo449) were compared to the radiosonde profiles, as well
as to the NN retrievals. BAO tower temperature and mixing ratio data at the seven available
levels were used as an additional validation dataset, without any interpolation.

To compare radiosonde observations against the PR and NN retrieved profiles, all these

profiles were interpolated vertically to the same PR heights, and PR and NN profiles were
averaged in the time window between 15 minutes before and 15 minutes after each
radiosonde launch. Since the radiosonde ascends quite quickly in the lowest kilometers of the
atmosphere (~15-20 min to reach 5 km), we estimated that the 30-minute temporal window is
representative of the same volume of the atmosphere measured by the radiosonde.

An example of the different temperature retrievals and their relative performance, data

obtained on 17 March 2015 at 2200 UTC is presented in Fig. 2. Temperature profiles up to 2 km
AGL from the four PR configurations (MWRz, MWRzo, MWRzo915, MWRzo449) are compared
to the radiosonde data in red, to the BAO measurements in blue squares, and to the NN profiles
(NN zenith in beige, and NN oblique in green). The MWRz and MWRzo profiles, as well as those
from the NNs, are very smooth and depart quite substantially from the radiosonde
measurements, being unable to reproduce the more detailed structure of the atmospheric
temperature profile measured by the radiosonde, while the MWRzo449 profile (in light-blue)
demonstrates better agreement with both the radiosonde and BAO measurements (blue
squares). Note that all four of the PRs match the BAO observations reasonably well, while the
NN retrievals are warm-biased.  The MWRzo915 profile (in magenta) also tries to follow the



elevated temperature inversion observed by the radiosonde, successfully only in the lower part
of the atmosphere (below 1 km AGL) where RASS 915 measurements are available. This
behavior will be also addressed in the following section and in the statistical analysis presented
later in the manuscript.

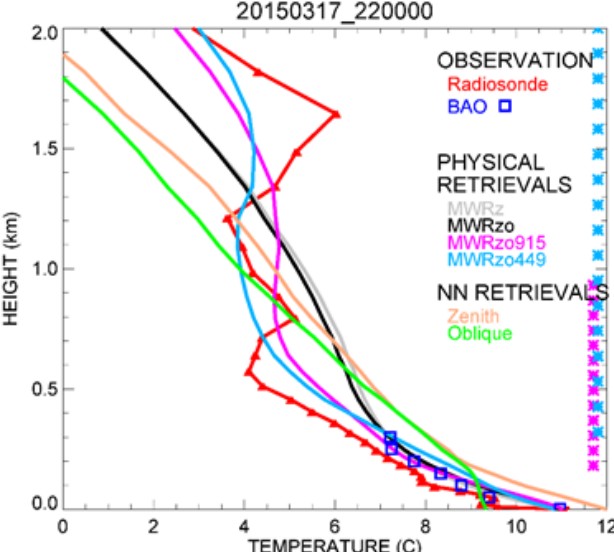


*Fig. 2. Temperature profiles obtained by the four PR configurations: MWRz in gray, MWRzo in*
*black, MWRzo915 in magenta, and MWRzo449 in light-blue; NN retrievals: NN zenith in beige,*
*and NN averaged oblique in green. These retrievals are compared to radiosonde measurements,*
*in red, and BAO tower observations, in blue squares. The heights with available RASS virtual*
*temperature measurements (RASS 915 in magenta and RASS 449 in light-blue), are marked by*
*the asterisks on the right Y-axis.*

**3.3 Averaging kernel**



The averaging kernel, **Akernel** (Masiello et al., 2012, Turner and Löhnert, 2014) from Eq.

(1) can be calculated as:

$$Akernel = B^{-1} K^T S_\varepsilon^{-1} K \qquad (2)$$

where:

$$B = S_a^{-1} + K^T S_\varepsilon^{-1} K$$

Both matrices, **Akernel** and **B**, have dimensions 111 x 111 in our configuration. The

**Akernel** matrix has  useful information about the calculated retrievals, such as vertical
resolution and degrees of freedom for signal at each level. Thus, the rows of **Akernel** provide
the smoothing functions that have to be applied to the retrievals (Rodgers, 2000) to help
minimize the vertical representativeness error in the comparison between the various retrievals
and the radiosonde profiles due to very different vertical resolutions of these profiles.

Using the averaging kernel, the smoothed radiosonde observed profiles will be

therefore computed as:

$$X_{smoothed\_sonde} = Akernel\,(X_{sonde} - X_a) + X_a \qquad (3)$$

The **Akernel** in Eq. (2) depends on the retrieval parameters (e.g., which datasets are

used in the **Y** vector, the values assumed in the observation covariance matrix $S_\varepsilon$, and the
sensitivity of the forward model (i.e., its Jacobian), etc.), so for our four PR configurations it is


possible to calculate four different kernels: **A_MWRz, A_MWRzo, A_MWRzo915 and**
**A_MWRzo449**, respectively.
While the top left corner of the **Akernel** matrix (1:55, 1:55) is devoted to temperature,
and it will be called **AT_MWR** hereafter, the next (56:110, 56:110) elements are devoted to
water vapor mixing ratio, and will be called **AQ_MWR**.
For each of the four **Akernels**, a smoothed radiosonde profile can be computed for each
radiosonde profile using Eq. (3). In the presence of temperature inversions or other particular
structures in the atmosphere these smoothed profiles can be quite different from each other
and also from the original unsmoothed radiosonde profile.
Therefore, in the statistical analysis presented later in the manuscript (in section 4.2),
mean bias, root mean square error (RMSE), and Pearson correlation coefficients will be
computed between the MWR's retrievals and both the unsmoothed and smoothed radiosonde
profiles,where the latter were computed using their respective **Akernels**.  Additional
observational data help to resolve the atmospheric structure in more detail, therefore we
would expect to obtain better statistical evaluations from the configurations including
additional RASS observations compared to the runs without RASS data.
The improvement in the retrieved temperature profiles presented in Fig. 2 obtained
using additional RASS data can be explained and clearly shown by the **ATkernel** itself. Figure 3
includes the temperature profiles of the radiosonde and PRs of MWRzo and MWRzo449 (panel
a), and the **ATkernels** corresponding to these PRs in the color plots in the middle of the figure



(panels b and c). These color plots are a schematic visualization of the 37 x 37 top left corner of
the **ATkernel** matrix that illustrates the part of the **ATkernel** up to 3 km, for reference. Dash
lines mark the 2 km vertical level.

The rows of the **ATkernel** provide a measure of the retrieval smoothing as a function of

altitude, so the full-width half maximum of each **ATkernel** row estimates the vertical resolution
of the retrieved solution at each vertical level (Merrelli and Turner, 2012). These plots of
temperature vertical resolution vs height for MWRzo and MWRzo449 are included in Figure 3,
panel d, for the same case presented in Fig. 2. Comparison of **ATkernel** color plots and vertical
resolution plots of MWRzo vs MWRzo449 shows that additional observations from the RASS
significantly reduces the spread around the main diagonal up to 2 km (in the layer of the
atmosphere where RASS 449 measurements are available), thereby improving the vertical
resolution of the retrievals (as clearly visible in panel d).

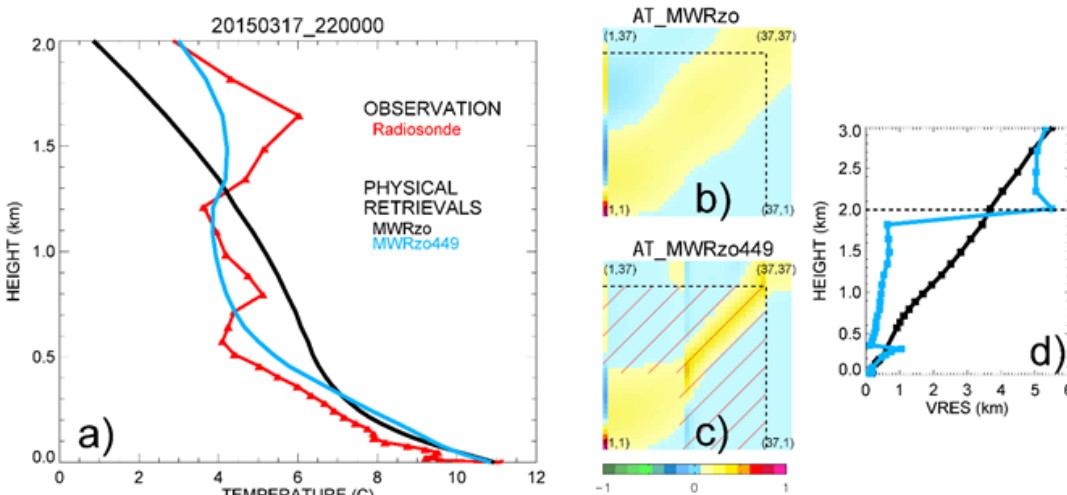




*Fig. 3. Panel a: temperature profiles from radiosonde, in red, from MWRzo PR in black, and from*
*MWRzo449 PR in light-blue. Middle colored panels: 37x37 levels (surface to 3 km) of the Akernel*
*matrix for temperature, b)* **AT_MWRzo** *and c)* **AT_MWRzo449***. Right panel d: vertical resolution*
*(VRES) as a function of the height for the MWRzo PR (black), and for the MWRzo449 PR (light-*
*blue). Dash lines on plots b)-d) mark 2 km AGL. Hatched area on panel c) marks the RASS*
*measurement heights.*

**4. Results**

441       PR and NN retrieved profiles have been evaluated against radiosonde observations. For

additional verification, radiosonde data from 59 launches taken between 9 March and 4 May
2015 were first of all compared to the BAO tower measurements, up to 300 m AGL. These
observed data sets match very well, with a correlation coefficient of 0.99 and a standard
deviation of ~0.7 °K.  However, one radiosonde profile showed a large bias (> 5 °K) against all
seven levels of BAO temperature measurements and against all PRs and NNs, therefore we
decided to exclude this particular radiosonde profile from the statistical calculations.

**4.1 PRs statistical analysis**

To complete the analyses on the **ATkernel** changes and dependencies from different

types of observational data used in the PRs, the **ATkernels**, averaged over all radiosonde
events, are shown in Fig. 4, panels a-d, for the four PR configurations of Table 1, in the same
way as shown in Fig. 3, b-c. A clearly visible gradual narrowing of the spread around the main
diagonal is obtained by the usage of the additional observations, from MWR zenith only (panel



a), to MWR zenith-oblique (panel b), to the larger impact obtained by the usage of RASS 915
(panel c) and RASS 449 (panel d) data.

Other statistically important features to analyze in the PRs, besides vertical resolution,

are the retrieval uncertainty, and the degree of freedom for signal (DFS). These three features
are also shown in Fig.4, panels e-g, at each of the heights of the retrieved solution, up to 3 km
AGL, and averaged over all radiosonde events. While the vertical resolution (panel e) shows the
width of the atmosphere layer used for each retrieval height (the vertical resolution is
computed as the full-width half-maximum value of the averaging kernel), the uncertainty (panel
f) gives a measure of the retrieval correctness (computed by propagating the uncertainty of the
observations and the sensitivity of the forward model), and the DFS (panel g) is a measure of
the number of independent pieces of information used in the retrieved solution. For example,
at the 1 km AGL level the vertical resolution of MWRzo449 equals 0.5 km, i.e. information from
+/- 0.5 km around the retrieval height are considered in the retrieval, while all other retrievals
use the information from +/- 2 km. Also, the uncertainty of the MWRzo449 retrieval up to 1km
AGL is around 0.5 $^{\circ}$K while the other retrievals have higher uncertainties of up to 1 $^{\circ}$K. The
higher accuracy of the MWRzo449 retrievals is because they use more observational
information compared to the other retrieval configurations.





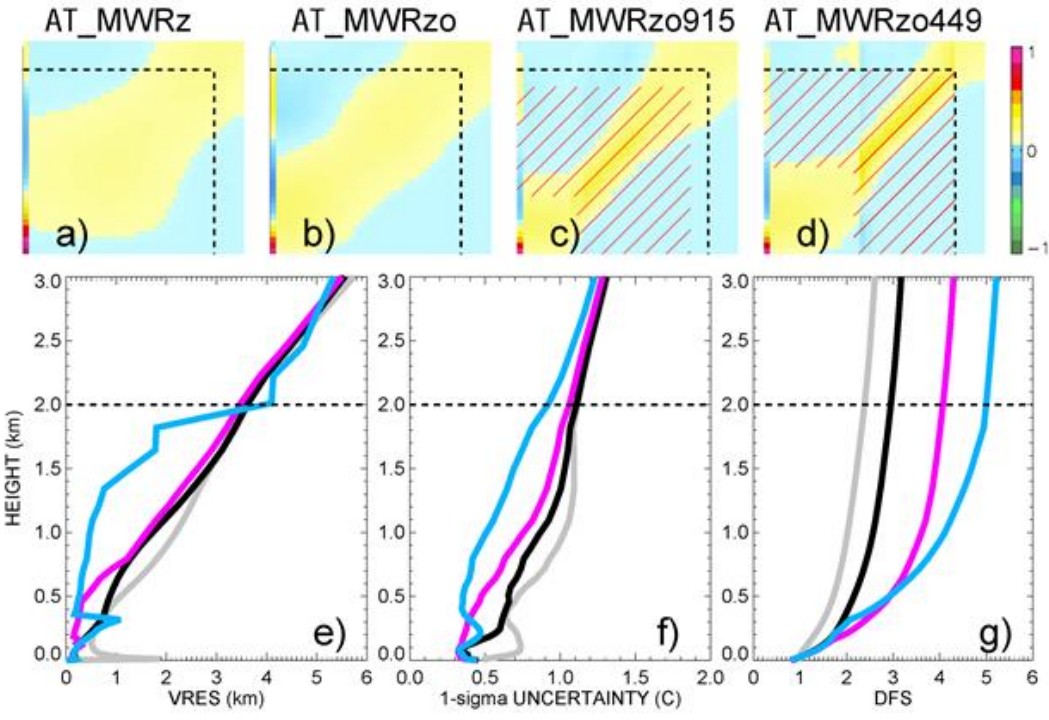


*Fig. 4. Top four-color images: **ATkernels** for MWRz (panel a), MWRzo (panel b), MWRzo915*

*(panel c) and MWRzo449 (panel d), averaged over all radiosonde events. Hatched area on*

*panels c) and d) marks the RASS measurement heights. Bottom three panels from left to right:*

*vertical resolution (VRES) in km (panel e), one-sigma uncertainty derived from the posterior*

*covariance matrix in ⁰C (panel f), and cumulative Degree of Freedom (DFS, panel g) as a function*

*of height for temperature, averaged over all radiosonde events (MWRz is in gray, MWRzo is in*

*black, MWRzo915 is in magenta, and MWRzo449 is in light-blue). Dash lines mark 2 km AGL on*

*all panels.*






The improvements from MWRz (in gray) to MWRzo (in black), then to MWRzo915 (in
magenta), and finally to MWRzo449 (in light-blue) are visible in all three panels (Fig 4 e-g),
whereas MWRzo449 has the best statistical measures compared to the other PRs, particularly
below 2 km AGL, where RASS 449 measurements are available. Finally, it is interesting that
below 200 m AGL the MWRzo915 has slightly better statistics compared to the MWRzo449, as
could be expected due to the first available height of the RASS 915 being lower (120 m AGL)
than the first available height for the RASS 449 (217 m AGL) and due to the finer vertical
resolution of the 915-MHz RASS. This suggests that if additional observations were available in
the lowest several 100 m layer of the atmosphere where RASS measurements are not available,
improvements might be even better closer to the surface, where temperature inversions, if
present, are sometimes difficult to retrieve correctly.
As a matter of fact, we found several cases during XPIA when the temperature profile
exhibits inversions, with the lowest happening in the surface layer. Figure 5a shows one of the
most complex cases, with several temperature inversions visible in the temperature profile
from the radiosonde (red line), in the temperature measurements from the BAO tower (blue
squares), and in the virtual temperature measured by the RASS 449 (light blue triangles). We
note that the virtual temperature profile is in close agreement with the temperature measured
by radiosonde. Generally, the moisture contribution to the virtual temperature is less than a
degree K, decreasing substantially for dryer air. Among the PR profiles, the PRs including RASS
data show better agreement with the radiosonde in the atmospheric layer where RASS
measurements are available, as was already shown in Fig. 2 for a different date. Unfortunately,
this better performance is not visible below the first available RASS measurement, i.e. from the



surface up to ~200m AGL, where the PRs with additional RASS data have the largest positive
bias compared to both radiosonde and BAO data in this layer. We believe that the MWR data,
especially those from the oblique scans, in this case have a bias in the observed brightness
temperatures that propagates through the retrieval calculations, and including other
observational data is not enough to correct it in the layer between the surface data and the first
available RASS measurement.

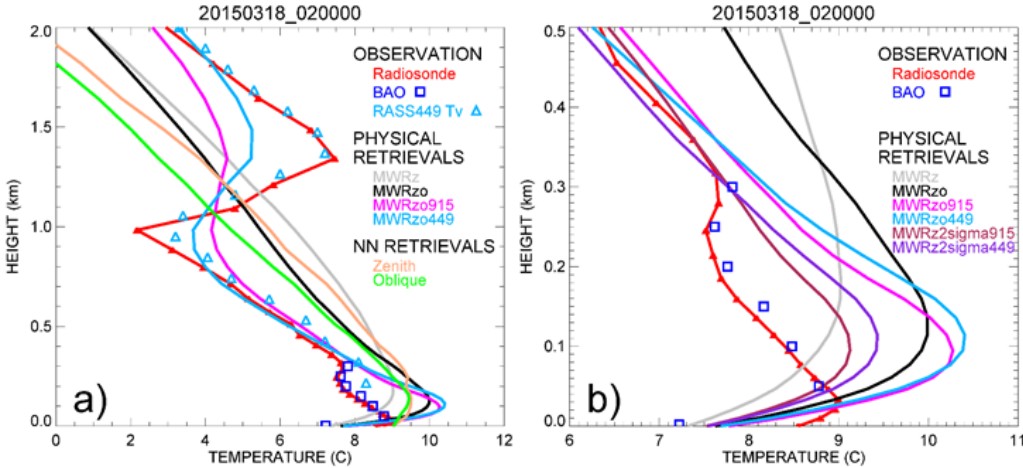


*Fig. 5. Panel a) as in Fig. 2 but for 18 March 2015 at 0200 UTC. The RASS 449 virtual*
*temperature is included as light blue triangles. Panel b) shows the same data (except for the NN*
*retrievals) presented in panel a), but only up to 500 m AGL, and includes PR profiles in which the*
*MWR uncertainties were increased by a factor of two, MWRz915 in maroon and MWRz449 in*
*violet.*





After several trials, we found that when RASS measurements are included, temperature

profiles in this and similar cases exhibiting inversions could be improved  by increasing the
random uncertainty of MWR observations, and only using the zenith MWR measurements,
because the oblique MWR brightness temperature measurements (which give more
information in the lower layer of the atmosphere) seemingly have a bias that competes with
the active and more accurate measurements from the RASS and surface observations. In this
way, the PR approach is granted more freedom to get an optimal profile in the gap between the
lowest RASS measurements and the surface measurement. Proof of this is presented in Figure
5b, that shows the same data as in 5a, but including the profiles obtained when increasing the
assumed MWR Tb uncertainties by a factor of two, hereafter called MWRz2sigma915 and
MWRz2sigma449, in maroon and violet respectively. The increased accuracy of these
temperature profiles compared to MWRzo915 and MWRzo449 are obvious in the layer of
atmosphere closer to the surface. Later we will show that these last two PR configurations
demonstrate improved statistics over all 58 cases, and also through the layer of the atmosphere
up to 5km. We note that these last two PR configurations, that were found to work well for this
dataset, might not be optimal for other datasets. During XPIA the RASS measurements impact
(particularly those from the RASS 449) was important in the PR approach. This might not be the
case for other datasets or over different seasons, when RASS coverage might not be as good as
that during XPIA. For this reason, we think that attention has to be used to determine what is
the best configuration to use when dealing with PR approaches. On the positive side, the
advantage is that the user can determine and has control on what is the optimal configuration
to use in his/her dataset, in terms of different inputs to employ and their relative uncertainty.




**4.2 Statistical analysis of PRs compared to NN retrievals**

Since the iteratively calculated PRs and the NN retrievals are obtained by very different
approaches, we find it very important to compare their relative statistical behavior. We do this
both for temperature and mixing ratio, providing this comparison in two ways: first using the
**Akernel** smoothed radiosonde data obtained as described in section 3.3, and second comparing
to the original, unsmoothed, radiosonde profiles, just interpolated to the 55 PR vertical levels.
Figure 6 shows the statistical results of these comparisons for temperature, in terms of
Pearson correlation, RMSE, and mean bias, averaged over all radiosonde events.

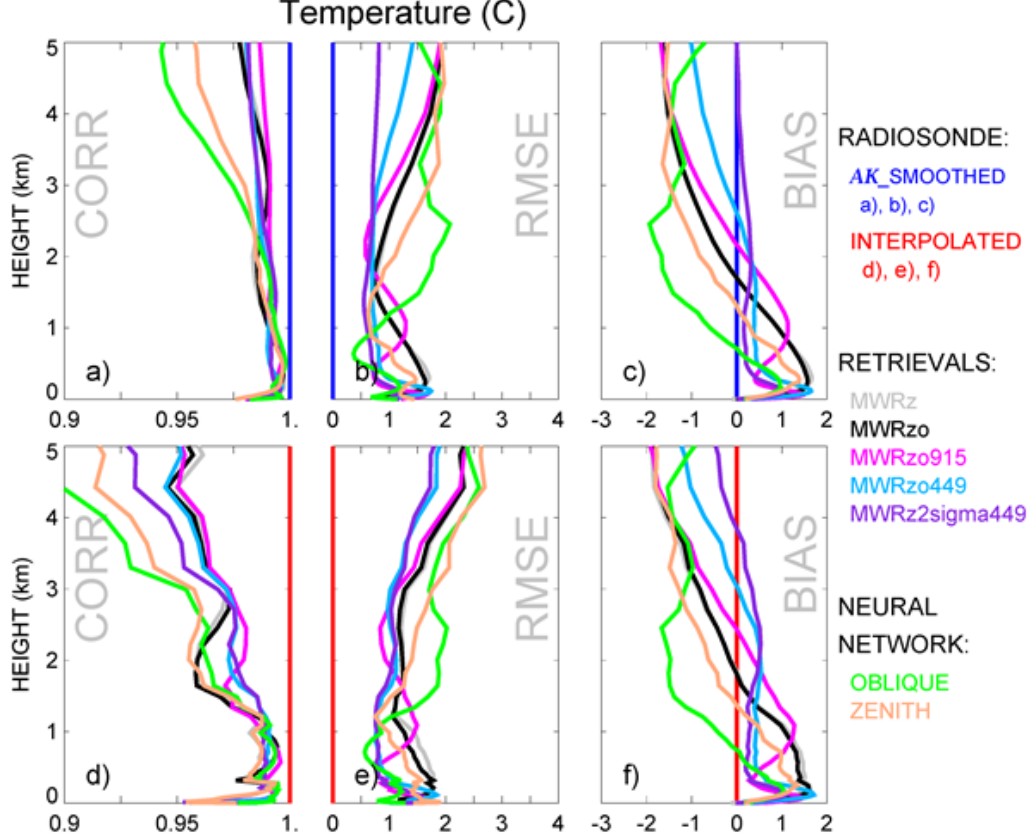


Fig. 6. Pearson correlation, RMSE, and mean bias for temperature profiles of MWRz in gray,

MWRzo in black, MWRzo915 in magenta, MWRzo449 in light-blue and MWRz2sigma449 in

violet, computed comparing to smoothed radiosonde data (using their relative **ATkernel**) in

panels a-c, and against the original radiosonde measurements in panels d-f. The same

comparisons for NN profiles, with NN zenith in beige, and NN averaged oblique in green, are

made against the corresponded smoothed radiosonde data in the top panel and against original

radiosonde data in the bottom panel.




These results confirm the superiority of the MWRz2sigma449 temperature retrieval

over the other PRs. While this is not true at all heights, this retrieval shows improved
distribution of RMSE and bias for the atmospheric layer up to 5 km AGL. The MWRz2sigma915
profile is not included in the figure to not overcrowd it, but its behaviour compared to the
MWRzo915 is similar to that of the MWRz2sigma449 compared to the MWRzo449 profile,
reducing the drastic bias found in the layer closer to the ground. The differences between the
two ways of comparison, against the smoothed **ATkernel** or the original radiosonde data, are
small in terms of RMSE and bias, but more evident in terms of correlation as it can be expected
because of the smoothing technique applied to the radiosonde profiles through Eq. (3). Above
and below 1.5 km AGL the bias, RMSE, and correlation profiles of the PRs show very different
behavior. While statistical measures above 1.5 km AGL are very similar for the four PRs
introduced in Table 1, they are better for the MWRz2sigma449 PR, especially when compared
to the smoothed radiosonde profiles. Differences between the profiles show more variability in
the lowest 1.5 km. NN retrievals, both for zenith and averaged oblique, are very variable from
height to height and generally have much larger RMSE and bias, and worse correlation
coefficients compared to PRs.

Besides temperature profiles, the NN and PR retrievals also provide water vapor mixing

ratio profiles.  It is understandable that the different configurations of PRs are not noticeably
different from each other in relation to moisture, because the Tv observations from the RASS
are dominated by the ambient temperature (not moisture), and thus have little impact on the
water vapor retrievals.





Figure 7 includes two **AQkernels** corresponding to the PRs MWRz and MWRzo449 in

panels a and b, which are averaged over all radiosonde events and appear to be almost
identical. More detailed statistical estimations of PRs mixing ratio in Fig 7 c-e, also averaged
through all radiosonde events, show very similar correlations, RMSEs, and biases for all PRs
included in the figure, meaning that the impact of including RASS observations is minimal on
this variable. These PR mixing ratio profiles are also statistically very close to the averaged
oblique NN retrieval mixing ratio profiles, with the zenith NN retrieval mixing ratio profiles
showing the worst statistics in terms of RMSE and bias. Overall, we conclude that the PR
retrievals are not degraded on average compared to the NN moisture retrievals.





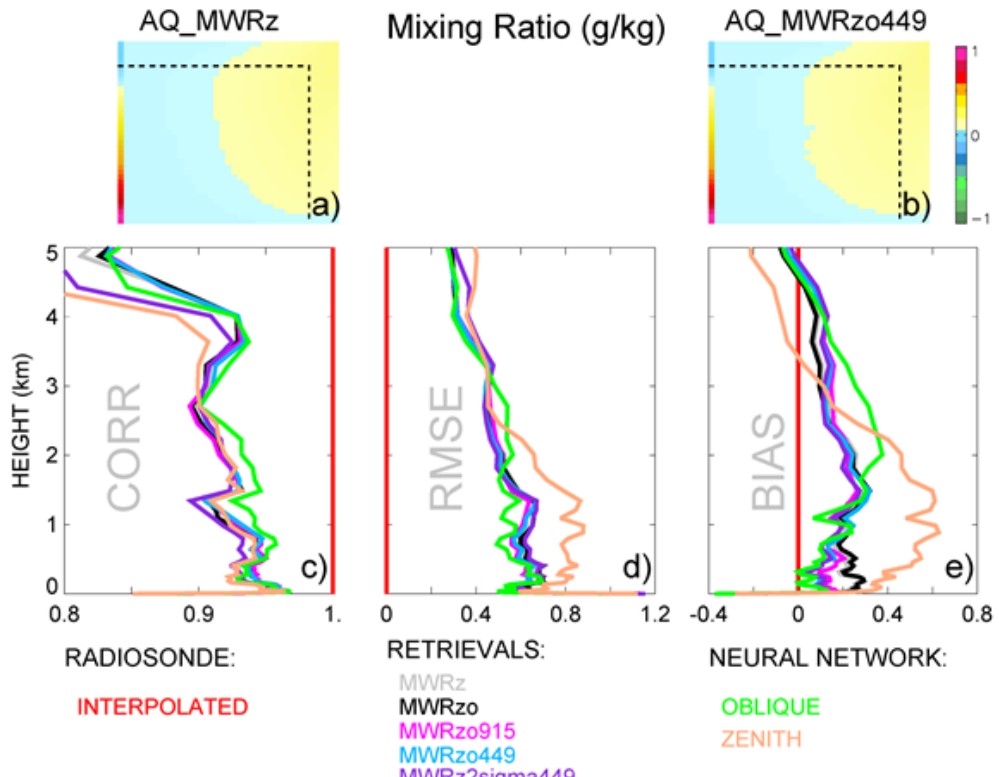


*Fig. 7. Top two-color images: **AQkernels** for MWRz (panel a) and MWRzo449 (panel b),*

*averaged over all radiosonde events and shown up to 3 km AGL with dash lines mark 2 km AGL*

*on both panels. Bottom three panels are the same as panels d-f in Figure 6, but for mixing ratio*

*estimation.*

**4.3 Statistics for cases far from the climatological mean**

While both approaches, physical and neural network retrievals, are quite different, both

use climatological data as a constraint or for building the statistical relationships used in the

retrieval. Statistically, the averaged profiles of both temperature and moisture variables are





*600*   very close to the climatological averages. However, the most interesting and difficult profiles to

*601*   retrieve are the cases furthest from the climatology (Löhnert and Maier, 2012). To check the

*602*   behavior of the retrieved data in such events, we first calculated the RMSE for each radiosonde

*603*   profile relative to the prior profiles for 42 vertical levels from the surface up to 5 km AGL, and

*604*   then we selected the 15 cases with the largest 0-5km layer averaged RMSEs compared to the

*605*   prior. All comparisons are done against the corresponded smoothed **ATkernel** radiosonde data,

*606*   using **AT_MWRz, AT_MWRzo, AT_MWRzo915, AT_MWRzo449, AT_MWRz2sigma915,**

*607*   **AT_MWRz2sigma449** for all six PRs, and **AT_MWRz, AT_MWRzo** for NN zenith and NN oblique

*608*   retrievals respectively.

*609*

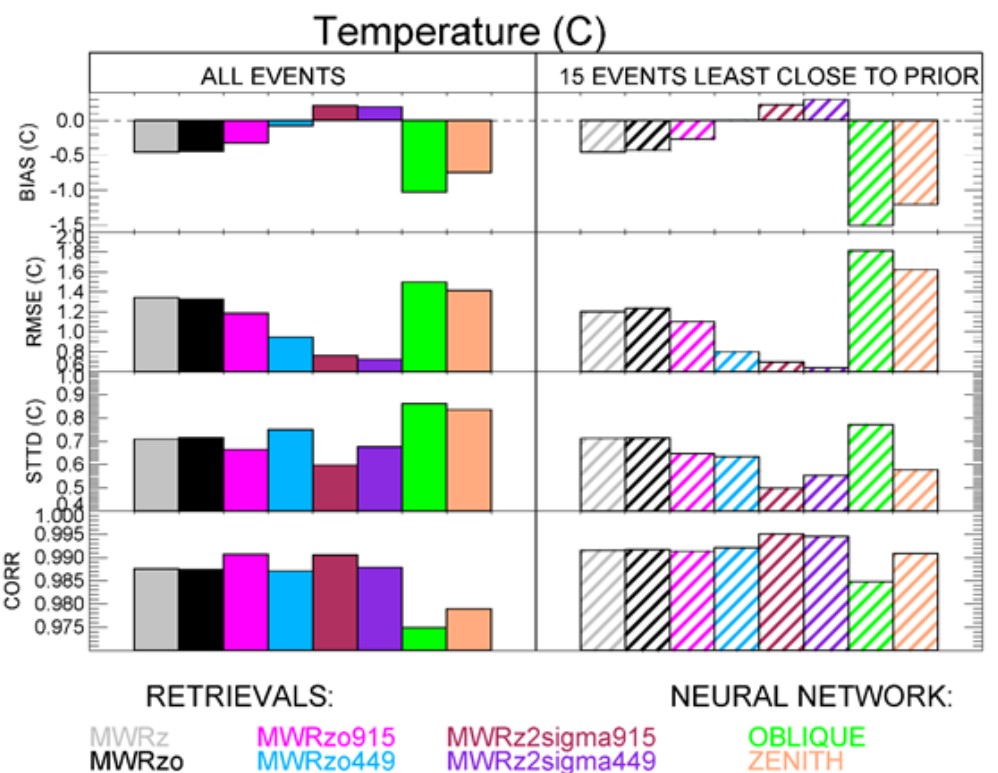

*610*



*611    Fig. 8. From top to bottom: biases (retrievals minus ATkernel radiosonde), RMSEs, standard*

*612    deviations of the difference between retrievals and ATkernel radiosonde, and Pearson*

*613    correlations for the six PR configurations so far introduced and both NN retrievals, averaged*

*614    from the surface to 5 km AGL, averaged over all radiosonde data (solid boxes), and averaged*

*615    over the 15 events furthest from the priors (hatched boxes).*


Figure 8 shows the temperature statistical analysis for the entire radiosonde data set

(solid boxes) and to just the fifteen chosen events (hatched boxes) for bias, RMSE, standard
deviation of retrieval differences to the radiosonde data, and Pearson correlation, calculated as
the weighted averaged over the 42 vertical heights up to 5 km AGL. Differences in the statistics
when using the entire radiosonde data set or the fifteen profiles furthest from the prior are
noticeable, especially for bias and RMSE, but also for the standard deviation. All PRs that
include RASS observations show better performance compared to strictly MWR-only PR profiles
(i.e., MWRz and MWRzo) for almost all statistical comparisons.  Also, the statistical behavior of
the MWRz2sigma915 and MWRz2sigma449 retrievals are the best in terms of RMSE and
standard deviation for all events and for RMSE, standard deviation, and correlation coefficient,
for the fifteen profiles furthest from the climatological average.  Finally, we note that the NN
profiles are the least accurate retrievals for all of the statistics for the entire radiosonde data
set, and have the highest bias, RMSE and the lowest correlation for the 15 events.

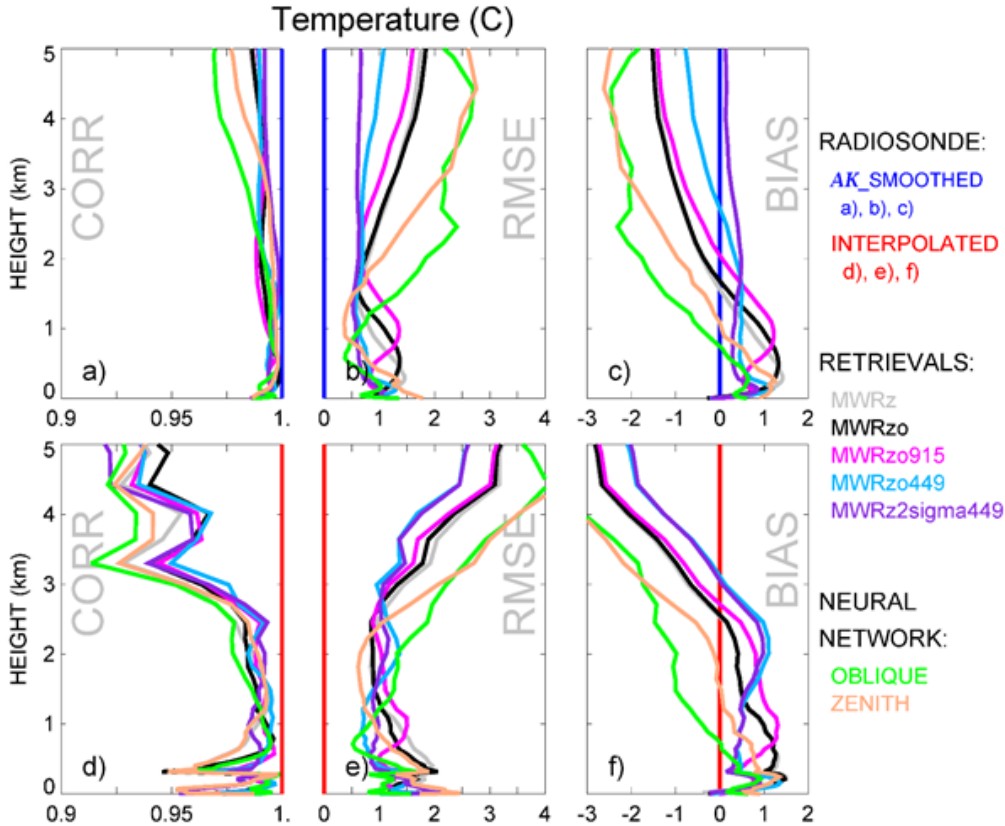


*Fig. 9. The same as Fig. 6 but for the temperature over 15 furthest from prior radiosonde*
*profiles.*

To investigate the vertical structure of the error statistics for the 15 events furthest from

the radiosonde climatology, profiles of correlation, RMSE and bias for these events are shown
in Figure 9 for the layer 0-5 km.  The MWRz449 and MWRz2sigma449 profiles, which were seen
in Fig. 8 to have the best layer averaged statistics, are seen to be as good as, or better, than the
other methods for the 0-2 km layer.  Importantly, for heights above 2km AGL, where there is no





additional observational data from RASS, all of the PRs are better than the NN profiles, with the
MWRz2sigma449 and MWRz449 being the best. We note that the increased accuracy of the
PRs relative to the NNs is more obvious in Fig. 9 for the 15 events when compared to the entire
data set in Fig. 6.  Also, it can be seen that the NNs for the 15 events are worse than they are
for the entire data set, especially in the 2-5km layer, which indicates (not surprisingly) that the
NNs accuracy degrades when the atmosphere is far from its climatology.

**4.4 Virtual temperature statistics**

The above analysis confirms the superiority of MWRz2sigma915 and MWRz2sigma449

compared to the other PRs and to the NN retrievals for this dataset. In this section we show the
direct comparison of the retrieved profiles to the original radiosonde and RASS virtual
temperature profiles. Using temperature and moisture retrieval output, we calculated
"retrieved virtual temperature profiles" and interpolated all profiles and RASS data on a regular
vertical grid, going from 200 m to 1.6 km with 100 m range, for easy comparison.

Figure 10 shows Tv retrieved profile biases compared to the original radiosonde data as

solid lines, and RASS 915 and RASS 449 Tv bias as asterisks. A zero bias is denoted by the red
line. On the left side of the figure we show bar charts of the RASS measurement availability as a
function of height. The widest part of these charts corresponds to 100% data availability.
Heights with RASS availability greater than 50% are marked with additional circles over the
asterisks.



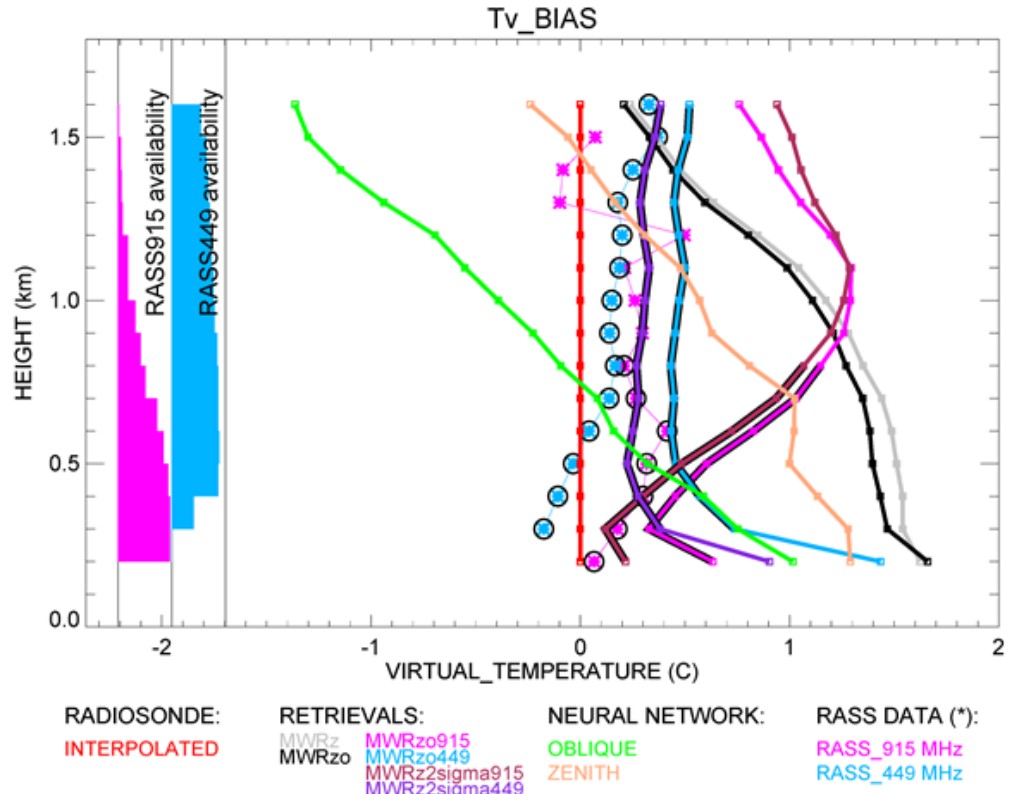

*659*

*660* *Fig. 10. Bias of virtual temperature for all six PR configurations and both NN retrievals*

*661* *compared to the original radiosonde measurements. RASS data are marked by asterisks and by*

*662* *additional circles for the RASS data with more than 50% availability, according to the availability*

*663* *bar charts on the left.*

*664*

*665*    While RASS 449 data are available at almost all heights up to 1.6 km, the RASS 915 data

*666*    availability decreases considerably with height, lowering to 50% availability around 800 m AGL.

*667*    All PRs with input from RASS data, MWRzo915 and MWRzo449, and MWRz2sigma915 and

*668*    MWRz2sigma449 with larger MWR uncertainties, are also marked with additional black lines at



the heights with at least 50% of relative RASS data availability. This figure clearly shows the
superiority of MWRz2sigma449 and MWRz2sigma915 (in the layer with > 50% RASS 915 data
availability) compared to MWRz and MWRzo configurations, which do not include RASS data, as
well as to MWRzo915 and MWRzo449 which include RASS data and MWR zenith and oblique
data. For MWRzo449 and MWRz2sigma449 profiles, RASS 449 data were almost always
available, therefore it is easy to identify similar features between Tv bias profiles of the RASS
449 and the PRs including it. Thus, for the MWRzo449 and MWRz2sigma449 the Tv bias is more
uniform through the heights compared to all other PRs that do not include RASS data, and to
both NN retrievals. Moreover, because MWRzo449 and MWRz2sigma449 Tv bias profiles follow
tightly the trend of the RASS 449 with height, the difference between MWRzo449 and RASS 449
biases equals ~0.32 °C and the difference between MWRz2sigma449 and RASS 449 biases
equals ~0.14 °C over the ~1.3 km atmospheric layer where RASS 449 measurements are
available, uniformly distributed through the heights. Finally, the average differences between
these MWRzo449 and MWRz2sigma449 Tv profiles and the radiosonde virtual temperature
equal ~0.56 °C and ~0.34 °C respectively. From these results we can assume that the final bias
of the PRs that include additional RASS data derives from a combination of the RASS data bias
itself, of the uncertainty of the retrieval model, and of the MWR brightness temperature biases,
even though we tried to correct for the latter.

We note as an alternative to using the PR temperatures at all heights, one could

consider replacing the PR temperatures with RASS observations up to the maximum height
reached by the RASS, and then use the PR retrieval above that.  To do this the moisture



contribution to the RASS virtual temperatures could be removed by using either the relative
humidity measured by radiometer or by a climatology of the moisture term.

**5. Conclusions**
In this study we used the data collected during the XPIA field campaign to test different
configurations of a physical-iterative retrieval (PR) approach in the determination of
temperature and humidity profiles from data collected by microwave radiometers, surface
sensors, and RASS measurements. We tested the accuracy of several PR configurations, two
that made use only of surface observations and MWR observed brightness temperature (zenith
only, MWRz, and zenith plus oblique, MWRzo), and others that included the active observations
available from two co-located RASS (one, RASS 915, associated with a 915-MHz, and the other,
RASS 449, associated with a 449-MHz wind profiling radar). Radiosonde launches were used for
verification of the retrieved profiles and Neural Network retrieved profiles were also used for
comparison. The NN retrievals used in this study were obtained either using the zenith angle
only, or the average of the oblique scans (based on the averaged Tb of 15- and 165-degree
scans) without including the zenith. Other MWR systems (Rose et al., 2005) provide retrieved
profiles that include the information from both oblique and zenith scans.

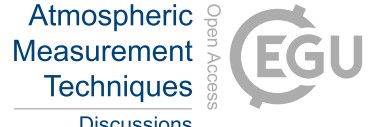
Inclusion of the observations from the active RASS instruments in the PR approach

improves the accuracy of the temperature profiles, particularly when low-level temperature
inversions are present. Of the PRs configurations tested, we find better statistical agreement
with the radiosonde observations when the RASS 449 is used together with the surface
observations and brightness temperature from only the zenith MWR observations
(MWRz2sigma449), and doubling the random radiometric uncertainty on the MWR
observations relative to the uncertainty calculated over the selected clear-sky days (Fig. 1). This
configuration is also more accurate compared to MWRzo915 or MWRz2sigma915 (which use
RASS 915 observation), because of the deeper RASS 449 height coverage.  The larger assumed
radiometric uncertainty in the MWR Tb observations allows the retrieval to overcome both (a)
the (small) systematic errors that exist between the MWR (which could be in either the
observed Tb values or in the MonoRTM used as the forward model) and the RASS, and (b) the
systematic errors that exist in forward microwave radiative models (Cimini et al. 2018).

We also selected 15 cases when temperature profiles from the radiosonde observations

were the furthest from the mean climatological average, and reproduced the statistical
comparison over this subset of cases. These are the cases usually most difficult to retrieve and
most important to forecast; therefore, it is essential to improve the retrievals in these
situations. Even for this subset of selected cases we find that MWRz2sigma449 produces better
statistics, proving that the inclusion of active sensor observations in MWR passive observations





would be beneficial for improving the accuracy of the retrieved temperature profiles also in the
upper layer of the atmosphere where RASS measurements are not available (at least up to 5 km
AGL).

Finally, we also considered the impact of the inclusion of RASS measurements on the

retrieved humidity profiles, but in this case the inclusion of RASS observations did not produce
significantly better results, compared to the configurations that do not include them. This was
not a surprise as RASS measures virtual temperature, effectively adding very little extra
information to the water vapor retrievals. In this case a better option would be to consider
adding other active remote sensors such as water vapor differential absorption lidars (DIALs) to
the PRs. Turner and Löhnert (2020) showed that including the partial profile of water vapor
observed by the DIAL substantially increases the information content in the combined water
vapor retrievals. Consequently, to improve both temperature and humidity retrievals a synergy
between MWR, RASS, and DIAL systems would likely be necessary.

**Data availability**

All data are publicly accessible at the DOE Atmosphere to Electrons Data Archive and

Portal, found at https://a2e.energy.gov/projects/xpia (Lundquist et al., 2016).

**Author contribution**



Irina Djalalova completed the primary analysis with physical retrieval approach through
MONORTM using XPIA data. Daniel Gottas contributed to the post-processing of the RASS data.
Irina Djalalova prepared the manuscript with contributions from all co-authors.

**Acknowledgements**
We thank all the people involved in XPIA for instrument deployment and maintenance,
data collection, and data quality control, and particularly the University of Colorado Boulder for
making the CU MWR data available. Funding for this study was provided by the NOAA/ESRL
Atmospheric Science for Renewable Energy (ASRE) program.

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
