# Peer review of "Improving thermodynamic profile retrievals from microwave radiometers by including Radio Acoustic Sounding System (RASS) observations Irina V. Djalalova1,2, David D. Turner3, Laura Bianco1,2, James M. Wilczak2, James Duncan1,2\*, Bianca Adler1,2 and Daniel Gottas2 1Cooperative Institute for Research in Environmental Sciences (CIRES), Boulder, CO, USA 2 National Oceanic and Atmospheric Administrati"

_Atmospheric Measurement Techniques, 2021_

## Referee Comment (RC1)

**Review AMT-2021-9:**

**Improving thermodynamic profile retrievals from microwave radiometers by including Radio Acoustic Sounding System (RASS) Observations**

**General comments :** The authors discuss about the improved accuracy of temperature profile retrievals obtained by combining active Radio Acoustic Sounding System (RASS) measurements and passive microwave radiometer (MWR) observations. This topic is of high interest for the scientific community as synergetic combinations of passive and active remote sensing instruments should help alleviate limitations when using each instrument alone. The structure of the manuscript is fine. However I found that the quality of English should be improved and the authors should be more precised in the presentation of their results. Several times, some details are missing to fully understand what they are discussing (the MWR configuration especially in terms of channels and elevation angles). I also think that several major scientific issues should also be addressed by the authors before publication to AMT. These issues concern the interpretation of the results using MWRs oblique measurements and some conclusions that, in my opinion, are not enough balanced due to the problems with the instrument used in this study. I also have some doubts for some interpretations for which additional explanations and potential new figures would be necessary to clarify the results. Thus I suggest the manuscript for publication after some major corrections in the results interpretation and minor english spelling errors which I list below.

**Specific comments :**

Major comment 1 : It should be clarified which MWR channels are used for each configuration : oblique versus zenith as well as for PR versus NN retrievals. In fact using transparent channels for lower elevation angles is often avoided as the homogeneity assumption is violated (especially if there is cloud or rain in one direction and not in the other direction when the two elevation scans are averaged). Thus, all my interpretation assumed that transparent channels are not used at low elevation angles for the manuscript. If this is not the case and transparent channels have also been used at low elevation angles, the authors should explicit which quality control has been used to identify inhomogenous scenes when they average the two microwave radiometer scans (refer to Cimini et al 2006).

Major comment linked with biases in MWR oblique scans :
The second major comment that should be addressed is about the interpretation of figure 5 where the degradation of the temperature profiles with MWRzo below 200m is attributed to biases in the MWR oblique scans. I think it is important to be rigorous there because nowadays many MWRs dedicated to temperature profiling use low elevation angles down to 5.4° to improve temperature retrievals. Thus, all your intepretation in the manuscript of the improvement brought by RASS measurements is sub-optimal if oblique scans cannot be used (at least for your conclusions below 2 km altitude where RASS brings most of the information). First of all, I think this needs to be addressed in the paper and clearly explained and discussed in the conclusion.
Secondly, I also found that the hypothesis provided in line 507 that your degraded results below 200 m with MWRzo comes from a bias is not convincing for several reasons :
→ line 207 : you mention that the two MWR units have a very good agreement in the temperature profiles in the overlapping dates both in terms of bias and correlation. Thus, when you conclude later that the MWR unit used in the paper presents a bias in the oblique measurements it means the two units were in fact biased and not well calibrated, which I found surprising (we can imagine a problem in one calibration but for two calibrations it seems that there is a problem in the deployment)
→ biases are in general very low for opaque channels that are the most informative below 1 km altitude (and even more below 200m where the degradation is observed). Liquid nitrogen calibration does not change so much the calibration for these channels as they are in general well

calibrated by the hot load calibration (every 5 minutes but I do not know if this is the case for the Radiometrics).

→ In figure 6, we can also see that NN retrievals using oblique measurements manage to improve NN with zenith only below 700 m, the degradation appears above 1 km when transparent channels are used and are more subject to large biases. Thus, the use of opaque channels below 700 m does not seem to degrade NN retrievals as much as shown for PR below 200m in figure 5. We observe the same thing in figure 2 : if we look at the NN retrievals, there is a significant modification of the profile below 250m when including oblique measurements that we do not observe with the physical retrievals.

In order to confirm your hypothesis, could you check the biases for oblique measurements as it is done in figure 1 ? If you compared to simulated TB from radiosondes and assuming homogeneity in an area around ~ 1km from the instrument, could you re-use the RS to investigate more in depth the biases at low elevation angles (as it is done in figure 1) to confirm this hypothesis ? Alternatively, you could also use model data (analysis or very short-term forecasts) during clear-sky conditions similarly to the paper of De Angelis et al 2017. I think this check is very important to confirm your conclusions lines 507 and 521. Depending on your answer about the channels used at oblique measurements, did you try to restrict MWRzo to only the most opaque channels (very close to 58 GHz) ? It would be interesting to identify if the supposed bias occurs for all V-band channels and/or only the most transparent ones.

Second major comment about NN retrievals:
Line 347 you mention that you cannot un-bias the BT from neural network. I can understand especially if you did not train the neural network by yourself but I think this is a major concern in all your evaluation of the next sections. We can see that NN retrievals have a degraded accuracy due to an increase bias above 1 km altitude which is probably due to the large V-band bias for transparent channels. However, after this small remark line 347, you never discuss this issue again. I think it is not fair when you compare with the PR which takes into-account a bias-correction which is very large for transparent V-band channels. At minimum, the authors should always remind this limitation to the reader : the problem might not be due to the NN approach itself but to a bias-correction that needs to be applied to NN retrievals similarly to PRs (you should also cite Martinet et al, Tellus, 2015 which shows how NN bias can be decreased after bias correction).
I am also wondering if, through the manufacturer software, you could re-process the NN retrievals by modifying the binary of TB files including the bias that you provided in figure1. This should be feasible and at least would give some ideas if the NNs are improved when using the same BT as for PRs (but keeping in mind that your bias correction for NN would not be perfect as probably a different RTM has been used to deduce the bias and train the NNs).
I also only understood at the end of the paper that the green line for the NN oblique measurements never use zenith observations. Thus, I assume NN with oblique measurements only does not use transparent channels as this would violate the homogeneity assumption. So, it is totally normal that the bias of NN with oblique measurements is degraded above 1 km altitude…If NN with oblique measurements only use opaque channels at low elevation angles, all your results to compare with NN retrievals should combine the two temperature profiles that you obtain: the one from zenith only mainly above 1 km altitude and the one obtain from oblique measurements below 1 km altitude. This has to be done if you want to compare with the configuration MWRzo which uses both zenith and oblique measurements. If I also understood correctly that zenith observations are not used for NN retrievals I think that figure 8 should stop at 1 km above ground maximum and not 5 km. Either you want to go up to 5 km altitude and you need to create a composite temperature profiles from the NN retrievals and make again your statistics with this new profile. Or you should limit your averaging of the bias and RMSE up to 1 km altitude because you cannot take into account statistics from the NN which are biased because they do not use observations informative of higher altitudes (or observations which are not bias corrected like the PRs).

**Technical corrections :**

Introduction, line 109 : I think the sentence is a bit too long and complex to follow. The radiative transfer equations are in general used to train the neural network retrievals or used directly inside physically-based retrievals whereas from the sentence it seems not connected. I think the sentence would be more rigourous rephrased that way :
« in order to estimate profiles of temperature and humidity from observ**ed** brightness temperatures, they apply regressions, neural network retrievals or physical retrieval methodologies which include more information about the atmospheric state in the retrieval process. Radiative transfer equations are commonly used to train statistical retrievals or as forward models inside physical methods».

Introduction line 116 : I do not agree with the argument that MWRs have a limited accuracy due to the fact that they do not actively measure temperature and humidity profile. We can of course improve their retrievals but it is hard to find sensors with accuracy better than 0.5 to 1.5 K during all conditions for temperature. I agree with the other drawbacks (lower accuracy during rain, coarse vertical resolution especially) but not with that one or you should give more arguments.

Introduction line 121 : site specific climatology is only a disadvantage for regressions or neural networks. This is not the case when using 1D-Var retrievals combined with model outputs. I think it would worth mentioning a few reference papers using 1D-Var approaches combined with NWP model : Hewison 2007, Cimini 2011, Martinet et al 2020 etc..

Introduction line 125 : The litterature refers more to low accuracy of MWR LWP retrievals for values below 20 g/m², 50g/m² seems a bit overestimated please modify or provide a reference for this statement.

Introduction line 142 : add an « s » to lowest several km**s**.

Section 2, line 172 : change included into including.

Section 2, line 196 : change manufacturing into manufacturer.

Section 2.1, line 203 : Please correct into : « NN zenith and of the NN oblique **measurements.** »

Section 2.1, line 205 : can you mention the date of the last calibration with liquid nitrogen for the data used in the paper ?

Section 2.2, line 221 : can you mention in which conditions RS were launched (how many clear-sky or cloudy-sky?)

Secion 2.3, line 225 : Please correct same location **as** the MWR.

Section 3.1, line 270 : please specify : integrated content of **liquid** water

Section 3.1, line 282 : could you add some spaces between the Sa matrice and the specification of the Jacobian Kij ? Could you also specify in this notation what is i and j ? (I assume channel and vertical level). Could you be consistent with the definition of Xa line 267 (always use L for LWP or only LWP everywhere) ?

Section 3.1, line 294 : can you say a word on how the Sa matrix has been computed ?

Section 3.1, line 296 : can you mention the perturbation size that you used to compute your jacobians ?

Section 3.1, line 300 : could you please mention which MWR channels are used in the retrievals for zenith only and for oblique measurement ? (all of them or just a sub-sample .).

Section 3.1, line 312 : could you mention the uncertainty values used in the Se matrix ?

Section 3.1 and table 1 : Does Tbzenith-oblique means both TB measured at zenith and at oblique elevation angles ? If this is the case, why there is a cross at the column indexed « Tbzenith » too ? It is a bit confusing as it seems that Tbzenith is used twice in the retrievals which I assume is not the case. Could you clarify this point in table 1 but also line 283 in the Se matrix ?

Section 3.1, line 278 : the sentence is confusing. It seems equation (1) is here to show how the Y vector is estimated from the state vector X whereas equation (1) shows the new atmospheric state updated at each iteration of the minimization depending on the previous state, the different matrices (Sa, K, Se) and the forward model. Please correct the sentence accordingly so that it makes more sense.

Section 3.1, line 313 : please correct the sentence into : « its dimensi**on** increas**es** ».

Section 3.2, line 319 : please correct into « will contribute to **a** bias **in** the retrievals ».

Section 3.2, line 328 : could you mention what thresholds and criteria you used from the 30 GHz Tb to identify clear-sky periods ? (standard deviation over which time period and which threshold?)

Section 3.2, line 333 : How the bias is computed ? Is it a difference with simulated BT from radiosondes ? Can you please clarify this in the manuscript.

Section 3.2, line 345 : can you at least mention that NN biases could be improved by applying a bias-correction ?.

Section 3.2, figure 2 : Can you specify if it is a clear-sky day or a cloudy day ? I suspect that this is a cloudy day with elevated inversion which often causes trouble to MWRs. If possible, a comparison with a clear-sky day by night with a sharp temperature inversion close to the surface could be interesting too. Could you say a word in the manuscript why you are have 0.5 to 1K difference between the RS measurements and the BAO tower measurements  which are used a the « truth » for validation ?

Section 3.2 line 366 : Modify the sentence into « demonstrate **a** better agreement ».

Section 3.3, line 388 : please rephrase into « Akernal **provides** useful information ».

Section 3.3, line 425 : please correct vs into versus.

Section 3.3, figure 3 : As it is, Panel a) does not sound really relevant to me as it is the same as figure 2. However, in this section we would expect  to see the smoothed RS profiles for the two configurations selected (MWRzo and MWRzo449). Could it be added to panel a) ?
Can you also explain why you get a strange vertical line in the Atkernel on the left part of the figure ?

Section 3.3, line 437 : change dash lines into dash**ed** lines.

Section 4.1, line 468 : to be consistent add a space to 1km => 1 km

Section 3.3, figure4 : can you explain why MWRzo915 does not make any improvement of the vertical resolution above ~600m compared to the MWRzo ? From panel c) it seems the spread around the diagonal is significantly reduced compared to MWRzo. However, the black and purple lines are almost on top of each other in panel e).

Section 4.1, line 479 : change dash lines into dashed lines.

Section 4.1, line 531 : add a space to « 5 km » to be consistent through the manuscript.

Section 4.1, line 535 : change as good as that during XPIA into « as good as during XPIA ».

Section 4.2, line 544 : please changed into « smoothed radiosonde **using the averaging kernel matrix »**.

Section 4.2, line 566 : change « above and below 1.5 km » into « by up to 5 km AGL »
.
Section 4.2, line 567 : change statistical measures into statistical scores.

Section 4.2, line 567 : I do not understand this sentence which is in contradiction with the previous one. Line 566 you mentin that statistical scores are very different for all PRs but then line 567 that above 1.5 km AGL they are similar. What do you mean ? Please correct the text accordingly.

Section 4.2, line 570 : Please change « NN retrievals are very variable » into « the accuracy of NN retrievals is very variable ».

Section 4.2, line 571 : Your conclusion is only true above 1 km altitude, below 1 km altitude, NN retrievals perform better than MWRz and MWRzo and even the two configurations with RASS measurements. The degradation of NN retrievals above 1 km is mainly due to a large bias which might be due to the fact that you do not apply the bias correction to MWR measurements for NNs whereas you apply it to the PRs. This needs to be justified and clearly stated here. Linked to my previous comment, I do not understand how NN retrievals can be improved below 1 km with oblique measurements whereas you concluded in section 4.1 that oblique measurements present a large bias. Additionally, the MWRz using only zenith measurements also present a large bias (above 1 K) below 1 km altitude which seems to conclude that probably opaque channels are biased both at zenith and oblique measurements. Could you also comment on the degradation of the accuracy of MWRzo915 between ~ 200 m and 1 km ? In figure 5 you showed an example were the RASS 915 measurements were able to improve temperature retrievals of MWRz and MWRzo above 200m but averaged over all the profiles it is not the case any more. It seems to come from a bias in your retrieval that we do not observe with MWRzo 449.

Section 4.2, figure 6 : I think the vertical blue and red lines to identify a correlation of « 1 », perfect RMSE of 0 and bias of zero are confusing for me. The figure being already crowded, I would remove these additional coloured lines for only a vertical black dashed line for panels c and f only.

Section 4.2, figure 7 : I am surprised that you use oblique measurements from the MWR for humidity retrievals : can you comment on the fact that this probably violates the homogeneity assumption necessary to use low elevation angles ? In general only opaque channels are used at low elevation angles and they are not sensitive to water vapor. If you used low elevation angles, did you apply a quality-control to detect inhomogenous cloudy scenes ?

Section 4.3, line 120 : what do you mean by « weighted average  over the 42 vertical heights » ?

Section 4.3, figure 8 : Could you comment about the potential modifications to your figure if you had calculated the statistics up to 1 km or 2 km AGL instead of 5 km ?
As you do not apply a bias correction to the NN retrievals where V-band transparent channels have a strong bias I am wondering if the conclusions are not wrongly biased for the evaluation of NN retrievals with this averaging up to 5 km. As already mentionned previously, it is not fair to compare two retrievals not applied on the same dataset (one with bias correction, another one without). At least this should be again commented when discussing the results of this section.

Conclusion line 703 : I honnestly did not have understand that NN retrievals with oblique measurements do not use the zenith observation. This has to be more explicit directly in section 2.1, line189 to 201. This explanation arrives too late in the manuscript

Conclusion : line 718 when MonoRTM is mentioned is redundant with line 719. I suggest modifying lines 717 to 719 into :
the small systematic errors that exist between the MWR observed Tb values and the RASS measurements and (b) the systematic errors that exist in forward microwave radiative models.
(I would thus remove all the text between parenthesis).

Conclusion, line 722 : please correct **the** most difficult to retrieve and **the** most important to forecast.

Conclusion, line 728: this sentence should be mitigated : the study proves that active sensors can improve MWR passive observations with zenith observations only but due to the weird results you obtain with lower elevation angles which are expected to improve the retrievals in the same area as the RASS measurements I think you should mention that the results could be different with MWRs with elevation angles usable down to 5° above the ground. In fact, with new MWR instruments using both zenith and low elevation angles we can expect RMSE between 0.5 and 1.5 K in the first 2 km (1.5 K for cloudy-scene when there is a temperature inversion in the upper layers). Thus, we cannot be sure that the improvement brought by RASS measurements would be as much informative in the first 1 km with a MWR unit for which oblique measurements could be optimally used. I think you should mention this in your conclusion.

---

## Referee Comment (RC2)

**Reviewer Comment** *on Improving thermodynamic profile retrievals from microwave radiometers by including Radio Acoustic Sounding System (RASS) observations* by **Djalalova et al.**

The submitted manuscript takes up on the ground-based remote sensing synergy approach of combining microwave radiometers (MWR) and RASS by applying a state-of-the-art physical retrieval approach. This is important, since MWR are known to show very accurate performance in temperature profiling in the lowest 500 m, whereas RASS are able to adequately capture the typical temperature inversion at the top of the atmospheric boundary layer (ABL) and thus, in theory, the synergy of both could lead to an improved temperature profile throughout the whole ABL.

**Major points**

1.) The way to showing the latter point above, however, is obviously severely hampered by the quality of the MWR data, most probably in terms of a TB bias. While the authors do show a bias correction applied to the MWR TBs, it is unclear whether this was done only for zenith observations or also at 15° elevation (Fig. 1). Here a detailed analysis is missing. If this manuscript is to be accepted for publication using real TB data, the reason for the biases shown in Figs. 6 c and f (black a grey lines) must be identified, discussed and corrected for.

2.) The paper shows hardly any quantitative discussion, which is necessary for a sound scientific analysis. Except for just a few passages, discussions of the figures are carried out only in a qualitative, rather unspecific manner. With respect to this, specifically the sections 3 and 4 should be thoroughly rewritten. E.g., avoid using "This might..", "We believe…", "seemingly", "Differences", "better" or "improve" etc. without referring to adequate statistical measures. A lot of the data is there in the XPIA data set und you can use to confirm, deny or to quantify your assumptions, respectively results.

3.) Because the authors write they could not apply any bias correction to the NN approaches, I strongly suggest omitting them from the paper. The comparisons are thus "unfair" and I do not see what benefit the reader has from including the NN retrievals when the actual goal is evaluating the MWR/RSS synergy potential that can be achieved with the PR. Instead, in all the corresponding figures, I would like to see the results of RSS-only PR, i.e. without including the MWR so the reader has an impression what these systems are capable of in a stand-alone manner.

4.) How did you deal with clouds, what about precipitation? Did you retrieve LWP simultaneously to temperature and water vapor? What influence do clouds have on the retrieval? I find no information about this throughout the manuscript.

5.) The sections describing microwave radiometry need more background and scientific accuracy.

**Further specific points and questions to be addressed**

1.) Abstract, last paragraph: It is not clear if the improvements described refer to the PR compared to the NN or the MWR+RASS combination compared to the MWR-only retrieval.

2.) Introduction: A description of the physical principle that allows temperature (& humidity) profiling (and LWP retrieval) from passive MWR observations is missing.

When doing so, please consider reformulating the advantages and disadvantages of the MWR retrieval methodology, because they are currently not scientifically sound. Be sure to differentiate how the frequency dependence and elevation angle dependence of TB can both lead to resolving the temperature profile in the vertical.

3.) Line 109: MWR don't "apply radiative transfer equations and neural network retrievals…" – please reformulate.

4.) Line 115: Please make clear what you mean with "deep layer of the atmosphere".

5.) Section 2.1, lines 203-204: The purpose of using observations at 15° elevation is not to "average out small scale horizontal inhomogeneities of the atmosphere" but to obtain TB observations at different optical depths.

6.) Section 3.1, lines 280-286: Why does the Y vector and the error covariance matrix contain both "zenith" and "zenith+oblique" components. If I understand correctly, you can choose to use only zenith observations and add the off-zenith (=oblique) TBs to improve the retrieval? So then should it not be "zenith" and "oblique"? Please clarify.

7.) Line 310: Do you mean the covariance between the uncertainties of the measurements?

8.) Section 3.2: There seems to be a non-consistent use of terminology. Please use "uncertainty" only in the sense of random uncertainty and distinguish it clearly from systematic offset (=bias).

9.) Lines 323-324: erroneous, please reformulate in a consistent manner

10.) Line 327: The 30 GHz channel is not predominantly water vapor, but liquid water sensitive.

11.) Lines 328-330: "The random uncertainty in brightness temperature was calculated as its standard deviation during clear sky times and for this channel is approximately 0.3 K": Why is this calculated standard deviation related to the TB uncertainty? Over what time window did you average? What about water variability in the atmosphere during the calculation time? Why actually did you calculate this standard deviation and where do you use it in the course of your study?

12.) Lines 332-333: How were the clear-sky days selected?

13.) Lines 333-334: How did you calculate the bias?

14.) Before line 358: a description and a quantitative discussion of the Sa and Se matrices applied needs to be given before going on describing retrieval results.

15.) Lines 425 and following, referring to Fig. 3: quantitative argumentation missing and VRES "jumps" in Fig. 3 are not discussed

16.) Section 4.1, lines 469-471: unspecific sentence, please reformulate

17.) Fig. 5: How many cases are used for the statistics, how many are clear-sky, how many are cloudy sky? How did you deal with cloudy cases in general?

18.) Fig. 8: Can you derive meaningful statistical measures such as RMSE from only 15 cases?

19.) Fig. 9: The MWRz2sigma449 performs best compared to the other retrievals. This retrieval relies on an increase in the MWR uncertainty, which was chosen in an arbitrary manner. This choice should be thoroughly justified and set into context with the performance of the 449-only retrievals which I would like to see (see "Major points" above).

20.) Section 4.4, lines 683-686: This sentence is formulated in a general, rather unspecific way and could be given without any of the studies conducted here.

**Technical comments**

1.) Figures are given in rather low resolution, a higher one would have been nice to be able to better interpret the results.

2.) Equation fonts appear in a non-standard, unorganized way.

3.) In general: please write K or °C, but not °K.

4.) Section 3.1, lines 280-286: Numerate all equations, be consistent with equation fonts and text fonts, be consistent with variables (i.e. L, LWP), explain all variables (and indices) in the text. Please be neater.

5.) Line 348 and following: use a new sub-section, the paragraphs are not related to "Bias-correction" anymore

6.) Section 3.3, lines 415-421: move text to Fig. 3 caption

---

## Author Response (AR1)

**Reviewer #1, answers.**

We wish to thank the reviewer very much for carefully reading our manuscript and for offering many comments towards its improvement. In revising the manuscript, we have taken into account almost all these comments.

Major comment 1 : It should be clarified which MWR channels are used for each configuration : oblique versus zenith as well as for PR versus NN retrievals. In fact using transparent channels for lower elevation angles is often avoided as the homogeneity assumption is violated (especially if there is cloud or rain in one direction and not in the other direction when the two elevation scans are averaged). Thus, all my interpretation assumed that transparent channels are not used at low elevation angles for the manuscript. If this is not the case and transparent channels have also been used at low elevation angles, the authors should explicit which quality control has been used to identify inhomogenous scenes when they average the two microwave radiometer scans (refer to Cimini et al 2006).

Two identical radiometers (Radiometrics MP-3000A) were used during the XPIA experiment. Both MWRs have 35-channels spanning a range of frequencies, with 21 channels in the lower (22-30 GHz) K-frequency band, from which 8 channels were used during XPIA: 22.234, 22.5, 23.034, 23.834, 25, 26.234, 28 and 30 GHz, and 14 channels in the higher (51-59 GHz) V-frequency band, all used in XPIA: 51.248, 51.76, 52.28, 52.804, 53.336, 53.848, 54.4, 54.94, 55.5, 56.02, 56.66, 57.288, 57.964 and 58.8 GHz, with elevation angles of 90 degrees (zenith) and 15 & 165 degrees (obliques). Section 2.1 has been modified to include these additions.

The Reviewer is correct in assuming that only the opaque channels are used from the oblique scans, when these are used in the Physical Retrieval approach. More specifically, the Physical Retrieval has two options for radiometer measurement inputs: using only the zenith scan, or using the zenith plus oblique averaged scans. From the zenith scan, Tbs from all 22 channels are used in both configurations, while for the oblique scans, when they are included, only the opaque channels (56.66, 57.288, 57.964 and 58.8 GHz) are used. Additional RASS active instrument measurements are used together with the second option, while 2-m in-situ observations of temperature and humidity are used in all configurations. So, Table 1 in manuscript has been revised to be:

|  | $T_{sfc}$ | $Q_{sfc}$ | $Tb_{zenith}$ | $Tb_{oblique\_avrg}$ | $TV_{RASS915}$ | $TV_{RASS449}$ |
|---|---|---|---|---|---|---|
| $Y_1 = MWRz$ | X | X | X |  |  |  |
| $Y_2 = MWRzo$ | X | X | X | X |  |  |
| $Y_3 = MWRzo915$ | X | X | X | X | X |  |

| $Y_4$ = MWRzo449 | X | X | X | X | | X |
|---|---|---|---|---|---|---|

The text has been modified in Section 3.1:

"The MWR provides **Tb** measurements from 22 channels from the zenith scan for the zenith only configuration (**$Y_1$,** which also includes the 2-m in-situ observations of temperature and humidity), while when using the zenith plus oblique Tb inputs (**$Y_2$, $Y_3$,** and **$Y_4$,** also including the 2-m in-situ observations of temperature and humidity) the same 22 channels were used from the zenith scan but only the four opaque channels (56.66, 57.288, 57.964 and 58.8 GHz) from the oblique scans."

**The second major comment** that should be addressed is about the interpretation of figure 5 where the degradation of the temperature profiles with MWRzo below 200m is attributed to biases in the MWR oblique scans. I think it is important to be rigorous there because nowadays many MWRs dedicated to temperature profiling use low elevation angles down to 5.4° to improve temperature retrievals. Thus, all your interpretation in the manuscript of the improvement brought by RASS measurements is sub-optimal if oblique scans cannot be used (at least for your conclusions below 2 km altitude where RASS brings most of the information). First of all, I think this needs to be addressed in the paper and clearly explained and discussed in the conclusion. Secondly, I also found that the hypothesis provided in line 507 that your degraded results below 200 m with MWRzo comes from a bias is not convincing for several reasons :

→ line 207 : you mention that the two MWR units have a very good agreement in the temperature profiles in the overlapping dates both in terms of bias and correlation. Thus, when you conclude later that the MWR unit used in the paper presents a bias in the oblique measurements it means the two units were in fact biased and not well calibrated, which I found surprising (we can imagine a problem in one calibration but for two calibrations it seems that there is a problem in the deployment)

The NN retrieved temperature profiles from the two MWRs indeed have a good agreement with statistically low bias (0.5 K) and high correlation (0.994). While in line 207 we refer to these statistical measures, line 507 refers to the particular case of March 18, 02:00 UTC which is certainly "a worst case scenario" in the XPIA experiment (certainly a difficult one to retrieve accurately from passive instruments because of the many temperature inversions, three in one profile including one at the surface!)

[Figure]

**Fig. R1.** Difference between bias-corrected Tb of MWR_CU and MWR_NOAA shown for all 34 days when both MWRs were available (left panel) and for the chosen four clear-sky days only (right panel). The averaged difference is shown in red. Blue lines Tb differences correspond to the rainy days.

Previously, in Bianco et al., 2017: "we compared the brightness temperatures (Tb) of the two MWRs for each retrieved channel **for 1 day**, finding almost all channels in good agreement (with differences of ∼ 2 K for channels 51.248, 51.760, 52.280, 56.020, 57.288, 57.964, and 58.800 GHz; differences of ∼ 1 K for channels 22.500, 26.234, 30.000, 52.804, 53.848, and 56.660 GHz; while the remaining channels did not show appreciable differences)". Using **all available data** from both radiometers we still found that the differences of their daily averaged Tb from the zenith scans, even bias-corrected, were 2-2.5 K for the opaque channels.

→ biases are in general very low for opaque channels that are the most informative below 1 km altitude (and even more below 200m where the degradation is observed). Liquid nitrogen calibration does not change so much the calibration for these channels as they are in general well calibrated by the hot load calibration (every 5 minutes but I do not know if this is the case for the Radiometrics).

Yes, the opaque channels have been changed less than the transparent ones by applying the bias-correction, so the initial difference between the two radiometers for these channels almost did not change. Additionally, the MWR_NOAA Tb had a problem with measurements in the 57.288 GHz channel (with an initial ~ 5 K difference with the MWR_CU Tb) but that difference was reduced in half with the bias-correction.

The Tb difference between the CU and NOAA radiometers for all available dates (34 days) after applying the bias-correction for the opaque channels (>56.5 GHz) still shows a difference around 2-2.5 K, while for the transparent channels (52-56 GHz) the differences are mostly improved with the final biases < 1.5 K. Finally, the K-band channel biases were not extremely large even without bias-correction, and after bias-correction they are less than a degree K different.

→ In figure 6, we can also see that NN retrievals using oblique measurements manage to improve NN with zenith only below 700 m, the degradation appears above 1 km when transparent channels are used and are more subject to large biases. Thus, the use of opaque channels below 700 m does not seem to degrade NN retrievals as much as shown for PR below 200m in figure 5. We observe the same thing in figure 2 : if we look at the NN retrievals, there is a significant modification of the profile below 250m when including oblique measurements that we do not observe with the physical retrievals.

In order to confirm your hypothesis, could you check the biases for oblique measurements as it is done in figure 1 ? If you compared to simulated TB from radiosondes and assuming homogeneity in an area around ~ 1km from the instrument, could you re-use the RS to investigate more in depth the biases at low elevation angles (as it is done in figure 1) to confirm this hypothesis ? Alternatively, you could also use model data (analysis or very short-term forecasts) during clear-sky conditions similarly to the paper of De Angelis et al 2017. I think this check is very important to confirm your conclusions lines 507 and 521. Depending on your answer about the channels used at oblique measurements, did you try to restrict MWRzo to only the most opaque channels (very close to 58 GHz)? It would be interesting to identify if the supposed bias occurs for all V-band channels and/or only the most transparent ones

Following the reviewer's suggestion, we compared the Tb measurements from the opaque channels for the same time shown in Fig.5 of the manuscript with Tb calculated by the forward model applied on the radiosonde data, Fig.R2 below:

[Figure]

**Fig.R2.** March 18, 2015, 02:00 UTC: the Tb of the opaque channels, 56.66, 57.288, 57.964 and 58.8 GHz, from the zenith scans (left panel) of the MWR_CU in red and MWR_NOAA in blue and from both oblique scans (right panel) in colors, and the Tb from the MonoRTM forward model using the corresponding radiosonde profile in black. Dashed color lines mark the original Tb data and solid color lines – bias-corrected Tb.

For this time period the bias-correction does not improve the Tb observed by the MWR compared to those derived from the radiosonde, the bias-corrected Tbs are further from the radiosonde Tbs compared to the uncorrected Tbs, except for the 57.288 GHz channel of the MWR_NOAA that shows measurement problems before bias correcting it. We have to admit that radiosonde Tb data cannot be claimed as the "true" because these data are the output of the forward model that has its own uncertainty.

The bias-correction in general improves the temperature profiles for most of the test time reducing the bias in the 1-2 km AGL layer by 0.5 K for all PR averaged profiles.

Additional text included in Section 3.2 (with some editing): "We compute the bias in the bias-correction procedure only from the zenith scans assuming that the same bias is suitable for the oblique scans. Also, we use the assumption that the true bias is an offset that is independent of the scene, so that the sensitivity to the scene (e.g., clear or cloudy, zenith or off-zenith) is small. To investigate this we eliminated the radiosondes launched during rainy periods (5 out of 58 cases) and found that the averaged temperature profiles were very little different than when all radiosonde profiles."

Fig.R2 shows the bias between the opaque channels' Tb and radiosonde-derived Tb. While these differences are similar in absolute values, but of opposite sign, for the zenith scans, the oblique channels show a noticeable difference between MWR_CU and MWR_NOAA Tbs compared to radiosonde Tbs (Fig.R2, right). These differences resulted in very different PR profiles from the two radiometers, as shown in Fig. R3:

[Figure]

**Fig.R3**. March 18, 2015, 02:00 UTC case. Observations from radiosonde are in red, and from BAO seven levels – in blue squares. The four PR profiles are in gray (MWRz), black (MWRzo), magenta (MWRzo915) and light-blue (MWRzo449). PRs from the MWR_CU are on the left and from the MWR_NOAA - on the right.

According to the right panel of Fig.R2, the MWR_CU has a bigger Tb bias for the opaque channels from the oblique scans compared to the MWR_NOAA measurements for this case, that resulted in MWR_NOAA temperature profile to be closer to the radiosonde profile in the layer of 0-300 m above the ground, shown in Fig.R3.

Still, with the measurement problem in the 57.288 GHz opaque channel of the MWR_NOAA instrument, and because of its limited time availability during the XPIA campaign, we decided to limit our analysis to the MWR_CU data only.

**Second major comment about NN retrievals**: Line 347 you mention that you cannot un-bias the BT from neural network. I can understand especially if you did not train the neural network by yourself but I think this is a major concern in all your evaluation of the next sections. We can see that NN retrievals have a degraded accuracy due to an increase bias above 1 km altitude which is probably due to the large V-band bias for transparent channels. However, after this small remark line 347, you never discuss this issue again. I think it is not fair when you compare with the PR which takes into-account a bias-correction which is very large for transparent V-band channels. At minimum, the authors should always remind this limitation to the reader : the problem might not be due to the NN approach itself but to a biascorrection that needs to be applied to NN retrievals similarly to PRs (you should also cite Martinet et al, Tellus, 2015 which shows how NN bias can be decreased after bias correction).

I am also wondering if, through the manufacturer software, you could re-process the NN retrievals by modifying the binary of TB files including the bias that you provided in figure1. This should be feasible and at least would give some ideas if the NNs are improved when using the same BT as for PRs (but keeping in mind that your bias correction for NN would not be perfect as probably a different RTM has been used to deduce the bias and train the NNs).

I also only understood at the end of the paper that the green line for the NN oblique measurements never use zenith observations. Thus, I assume NN with oblique measurements only does not use transparent channels as this would violate the homogeneity assumption. So, it is totally normal that the bias of NN with oblique measurements is degraded above 1 km altitude…If NN with oblique measurements only use opaque channels at low elevation angles, all your results to compare with NN retrievals should combine the two temperature profiles that you obtain: the one from zenith only mainly above 1 km altitude and the one obtain from oblique measurements below 1 km altitude. This has to be done if you want to compare with the configuration MWRzo which uses both zenith and oblique measurements. If I also understood correctly that zenith observations are not used for NN retrievals I think that figure 8 should stop at 1 km above ground maximum and not 5 km. Either you want to go up to 5 km altitude and you need to create a composite temperature profiles from the NN retrievals and make again your statistics with this new profile. Or you should limit your averaging of the bias and RMSE up to 1 km altitude because you cannot take into account statistics from the NN which are biased because they do not use observations informative of higher altitudes (or observations which are not bias corrected like the PRs).

We thank the referee for this particular comment. Following the very insistent recommendation of Reviewer #2 and your questionable opinion about the temperature comparison of bias-corrected input for the PR data and uncorrected NN data, we decided to move all comparisons of PR and NN profiles to Appendix A. The suggestion about NN bias-correction using the Tb biases from PRs looks interesting, but we decided not to mention it because of the artificial mix of two approaches. Instead, we included (in Appendix A) the comparison of PR profiles with separate NN profiles from zenith and from oblique averaged scans and with NN profiles calculated from the combination of the scans using NN oblique scans up to 1 km and NN zenith scans above.

**Technical corrections**:

Introduction, line 109 : I think the sentence is a bit too long and complex to follow. The radiative transfer equations are in general used to train the neural network retrievals or used directly inside physically-based retrievals whereas from the sentence it seems not connected. I think the sentence would be more rigourous rephrased that way :
« in order to estimate profiles of temperature and humidity from observed brightness temperatures, they apply regressions, neural network retrievals or physical retrieval methodologies which include more information about the atmospheric state in the retrieval process. Radiative transfer equations are commonly used to train statistical retrievals or as forward models inside physical methods».
Rephrased as suggested.

Introduction line 116 : I do not agree with the argument that MWRs have a limited accuracy due to the fact that they do not actively measure temperature and humidity profile. We can of course improve their retrievals but it is hard to find sensors with accuracy better than 0.5 to 1.5 K during all conditions for temperature. I agree with the other drawbacks (lower accuracy during rain, coarse vertical resolution especially) but not with that one or you should give more arguments.
We deleted the comment on the accuracy of temperature and humidity measurements.

Introduction line 121 : site specific climatology is only a disadvantage for regressions or neural networks. This is not the case when using 1D-Var retrievals combined with model outputs. I think it would worth mentioning a few reference papers using 1D-Var approaches combined with NWP model : Hewison 2007, Cimini 2011, Martinet et al 2020 etc..
We have now added the following text in the Introduction, together with the mentioned References: "Some studies have used analyses from NWP models as an additional constraint in these variational retrievals (e.g., Hewison 2007, Cimini 2011, Martinet et al. 2020); however, we have elected not to include model data in this study because we wanted to evaluate the impact of the RASS profiles on the retrievals from a purely observational perspective "

Introduction line 125 : The literature refers more to low accuracy of MWR LWP retrievals forvalues below 20 g/m², 50g/m² seems a bit overestimated please modify or provide a reference for
this statement.
Changed from 50 to 20.
Introduction line 142 : add an « s » to lowest several kms.

Included.

Section 2, line 172 : change included into including.
"Included" is right.

Section 2, line 196 : change manufacturing into manufacturer.
Changed.

Section 2.1, line 203 : Please correct into : « NN zenith and of the NN oblique measurements. »
Included.

Section 2.1, line 205 : can you mention the date of the last calibration with liquid nitrogen for the
data used in the paper ?
Prior to the experiment, both MWRs were calibrated using an external liquid nitrogen target and an internal ambient target and thoroughly serviced (sensor cleaning, radome replacement, etc.). The MWR used in this study was serviced and calibrated on 2/27/2015. This text was included in the manuscript.

Section 2.2, line 221 : can you mention in which conditions RS were launched (how many clear-sky
or cloudy-sky?)
Of 58 valid radiosonde profiles, 41 were launched in clear-sky periods, 12 - in cloudy periods, and 5 during rain. We defined those categories using Tb in the 30 GHz channel, as shown in the figure below:

[Figure]

**Fig.R4**. Zenith Tb from the 30 GHz channel for a clear-sky day (left panel), cloudy day (middle panel) and rainy day (right panel) from the CU radiometer in red and NOAA radiometer in blue. STDDEV(Tb-SMOOTH(tb,11)) is shown at the bottom of each panel with its average values printed under the panels in corresponding colors. Vertical lines (green – for clear-sky, beige – for clouds and cyan – for rain) show the time of radiosonde launches.
We also included the following text in the manuscript:

"Four clear-sky periods have been chosen using a criterion of less than 0.3 K uncertainty in the 30 GHz channel: March 10 and 30, and April 13 and 29, 2015. During periods with liquid-bearing clouds overhead, this criterion is markedly higher (more than 0.7 K) and much higher for the rainy periods (> 4 K). While those calculations were applied on a daily basis, it is important to mention that the days are not uniform in terms of cloudiness or rain. Therefore, we used the data for the 2-3 hours bracketing the time of radiosonde launches to determine to which category a particular radiosonde profile belongs, clear-sky, cloudy or rain. In this way, we found that from 58 radiosonde launches used in our statistical analysis, 41 belong to the clear-sky category, 12 - to cloudy but non-precipitating conditions and 5 - to rainy periods."

Section 2.3, line 225 : Please correct same location as the MWR.
Corrected.

Section 3.1, line 270 : please specify : integrated content of liquid water
Included.

Section 3.1, line 282 : could you add some spaces between the Sa matrice and the specification of
the Jacobian Kij ? Could you also specify in this notation what is i and j ? (I assume channel and
vertical level). Could you be consistent with the definition of Xa line 267 (always use L for LWP or
only LWP everywhere) ?
Xa and Sa are changed (from L to LWP). Jacobian is moved to form the straight-line definition.
Notations of "i" and "j" are included.

Section 3.1, line 294 : can you say a word on how the Sa matrix has been computed ?Section 3.1, line 296 : can you mention the perturbation size that you used to compute your Jacobeans ?
We included the additional description of the Sa matrix in the text in Section 3.1: "Using 3,000 radiosonde launched by the NWS in Denver, we interpolated each profile to the vertical grid used in the retrieval, after which we computed the covariance of temperature and temperature, temperature and humidity, and humidity and humidity for different levels."

Section 3.1, line 300 : could you please mention which MWR channels are used in the retrievals for
zenith only and for oblique measurement ? (all of them or just a sub-sample .).
As mentioned earlier, 22 channels were used from zenith measurement and 4 channels (opaque) – from oblique (included in Section 3.1).

Section 3.1, line 312 : could you mention the uncertainty values used in the Se matrix ?

The uncertainty in the MWR Tb observations was set to the standard deviation from a detrended time-series analysis for each channel during cloud-free periods. The derived uncertainties ranged from 0.3 K to 0.5 K in the 22 to 30 GHz channels, and 0.5 to 1.0 K in the 52 to 60 GHz channels. We assumed that there was no correlated error between the different MWR channels.

For the RASS, collocated RASS and radiosonde profiles were compared and the standard deviation of the differences in Tv were determined as a function of the radar's signal-to-noise ratio (SNR). This relationship resulted in uncertainties that ranged from 0.8 K at high SNR values to 1.5 K at low SNR values. Again, we assumed that there was no correlated error between different RASS heights.

These additions are also included in Section 3.1.

Section 3.1 and table 1 : Does Tbzenith-oblique means both TB measured at zenith and at oblique elevation angles ? If this is the case, why there is a cross at the column indexed « Tbzenith » too ? It is a bit confusing as it seems that Tbzenith is used twice in the retrievals which I assume is not the case. Could you clarify this point in table 1 but also line 283 in the Se matrix ?

Table 1 as well as observational vectors Y2, Y3 and Y4 and matrix Sε have been modified.

Table 1 with its modifications has already been shown above. Vectors Y2, Y3 and Y4 and matrix Sε are modified as follows:

$$Y_1 = \begin{bmatrix} T_{sfc} \\ Q_{sfc} \\ Tb_{zenith} \end{bmatrix} \qquad Y_2 = \begin{bmatrix} T_{sfc} \\ Q_{sfc} \\ Tb_{zenith+oblique\ avrg} \end{bmatrix}$$

$$Y_3 = \begin{bmatrix} T_{sfc} \\ Q_{sfc} \\ Tb_{zenith+oblique\ avrg} \\ Tv_{RASS915} \end{bmatrix} \qquad Y_4 = \begin{bmatrix} T_{sfc} \\ Q_{sfc} \\ Tb_{zenith+oblique\ avrg} \\ Tv_{RASS449} \end{bmatrix}$$

$$S_\varepsilon = \begin{bmatrix} \sigma^2_{T\,sfc} & 0 & 0 & 0 \\ 0 & \sigma^2_{Q\,sfc} & 0 & 0 \\ 0 & 0 & \sigma^2_{Tb_{zenith}}\,① \ or\ \sigma^2_{Tb_{zenith+oblique\ avrg}}\,② & 0 \\ 0 & 0 & 0 & \sigma^2_{Tv_{RASS915(449)}}\,③\ or\ ④ \end{bmatrix}$$

Section 3.1, line 278 : the sentence is confusing. It seems equation (1) is here to show how the Y

vector is estimated from the state vector X whereas equation (1) shows the new atmospheric state updated at each iteration of the minimization depending on the previous state, the different matrices (Sa, K, Se) and the forward model. Please correct the sentence accordingly so that it makes more sense.

Corrected in Section 3.1: "The MonoRTM model **F** is used as the forward model from the current state vector **X**, Eq. (1), and is then compared to the observation vector **Y,** iterating until the difference between **F(X)** and **Y** is small within a specified uncertainty."

Section 3.1, line 313 : please correct the sentence into : « its dimension increases ».
Done.

Section 3.2, line 319 : please correct into « will contribute to a bias in the retrievals ».
Done.

Section 3.2, line 328 : could you mention what thresholds and criteria you used from the 30 GHz Tb to identify clear-sky periods ? (standard deviation over which time period and which threshold?)

This text was added to Section 3.2 (with some editing):

"A threshold value of 0.3 K has been used for the uncertainty calculation. Fig. R5 (see below) shows one of the clear-sky days, March 10, 2015. The final uncertainty equals the average of the Tb standard deviation in a one-hour window sliding through all data points of a day. It also could be computed as the standard deviation of the difference between Tb and smoothed Tb to eliminate daily temperature variability. Finally, there is a "standard" set of uncertainties used as the high boundaries for Tb uncertainty per MWR channels calculated empirically in previous experiments."

"For the four chosen clear-sky days not only the daily uncertainties of **30 GHz Tb** were below 0.3 K, but all three sets of uncertainties described above were extremely similar with the averaged difference less than 0.05 K."

[Figure]

[Figure]

Fig. R5. Left: Tb from MWR_CU 30 GHz channel for March 10, 2015, one of the chosen clear-sky days. The standard deviation (at the bottom, in red) is calculated as the averaged standard deviation of Tb in a one-hour window sliding through all data points of the day. Right: MWR_CU uncertainty, computed as an average over four clear-sky days using a sliding window (in red), smooth function (in blue), and the before mentioned "standard" values (in black) for all 22 channels.

Section 3.2, line 333 : How the bias is computed ? Is it a difference with simulated BT from
radiosondes ? Can you please clarify this in the manuscript.
From the modified text in Section 3.2:
"The bias was computed for each of the 22 channels as the averaged difference between the Tb from the MWR zenith observations, and the forward model calculation applied to the prior, over these selected clear-sky days, and then subsequently removed from all of the MWR observations."

Section 3.2, line 345 : can you at least mention that NN biases could be improved by applying a
bias-correction ?.
We moved the NN and PR comparison in Appendix A and mentioned this possibility.

Section 3.2, figure 2 : Can you specify if it is a clear-sky day or a cloudy day ? I suspect that this is
a cloudy day with elevated inversion which often causes trouble to MWRs. If possible, a
comparison with a clear-sky day by night with a sharp temperature inversion close to the surface
could be interesting too. Could you say a word in the manuscript why you are have 0.5 to 1K

difference between the RS measurements and the BAO tower measurements which are used a the

« truth » for validation ?

[Figure]

**Fig.R6**. Oblique channels Tb from 30 GHz channel, March 17, 2015. Blue arrow marks the time of the day, 22:00 UTC, for the radiosonde case shown in Fig. 2 of the manuscript.

This is the 30 GHz channel Tb from the oblique scans. A difference between the two scans of 15 and 165 degrees at 22:00 (time of the radiosonde launch) just started to grow that may indicate the cloudiness in the view of one of the obliques.

Fig. 5 in the manuscript shows exactly one of the difficult cases you are mentioning: evening hours with sharp temperature inversions, one of them close to the surface.

[Figure]

**Fig. R7**. Averaged temperature at BAO tower heights from radiosonde (red) and from BAO levels (blue) in the left panel and their biases at each level with shaded image of standard deviation over 58 radiosonde launches in the right panel.

We indeed use BAO measurements as the "truth" having very close agreement between the radiosonde and BAO measurements. The special case of Fig. 2 in the manuscript has larger differences between the radiosonde and BAO, which on average were less than 0.5 K, which is within the expected accuracy of the radiosondes.

Section 3.2 line 366 : Modify the sentence into « demonstrate a better agreement ».
In the text now: "the MWRzo449 profile (in light-blue) demonstrates a better agreement"

Section 3.3, line 388 : please rephrase into « Akernal provides useful information ».
Done.

Section 3.3, line 425 : please correct vs into versus.
Included "versus".

Section 3.3, figure 3 : As it is, Panel a) does not sound really relevant to me as it is the same as
figure 2. However, in this section we would expect to see the smoothed RS profiles for the two
configurations selected (MWRzo and MWRzo449). Could it be added to panel a) ?
Can you also explain why you get a strange vertical line in the Atkernel on the left part of the
figure ?
The smoothed Radiosonde profiles from MWRzo and MWRzo449 are included in panel R8a).
First left vertical lines in panels R8b-c) indicate surface data (see the definitions of observational vectors Y). To confirm this, we repeated those runs without including surface temperature and humidity data in the observational vector. This indeed caused the disappearance of the vertical lines in Fig.R8b, c (not shown).

[Figure]

**Fig. R8.** The same as Fig.3 in the manuscript with two changes: T radiosonde profiles smoothed by AT_Kernel in MWRzo (dashed black) and MWRzo449 (dashed light-blue) are included in panel a), panel d) shows Vertical resolution calculated by FWHM method. These changes are included in the manuscript.

We also change the panel d) in this Figure by changing the method used to calculate the vertical resolution. There are two ways to compute the vertical resolution from the averaging kernel.

First, we applied a method that Tim Hewison published (TGRS 2007, reference below) that uses only the diagonal data of the averaged kernel. This method works well when the retrieval uses only the input from the passive observations, like the MWR, but is not very suitable for the passive/active combination of inputs, as was seen in Fig. 3d in the manuscript (with the creation of the "jumps"). So, we returned to the method (that we actually erroneously mentioned in the paper) that computes the vertical resolution as the full-width half-maximum (FWHM, TGRS 2008, reference below) value of the averaging kernel at each height.

T. J. Hewison, "1D-VAR Retrieval of Temperature and Humidity Profiles From a Ground-Based Microwave Radiometer," in IEEE Transactions on Geoscience and Remote Sensing, vol. 45, no. 7, pp. 2163-2168, July 2007, doi: 10.1109/TGRS.2007.898091.

Maddy, E. S. and C. D. Barnet, 2008: Vertical Resolution Estimates in Version 5 of AIRS Operational Retrievals. IEEE TGRS, VOL. **46**, NO. 8, AUGUST 2008, doi:10.1109/TGRS.2008.917498

Section 3.3, line 437 : change dash lines into dashed lines.
Changed.

Section 4.1, line 468 : to be consistent add a space to 1km => 1 km
Added.

Section 3.3, figure4 : can you explain why MWRzo915 does not make any improvement of the
vertical resolution above ~600m compared to the MWRzo ? From panel c) it seems the spread
around the diagonal is significantly reduced compared to MWRzo. However, the black and purple
lines are almost on top of each other in panel e).

The reason why the vertical resolution of the MWRzo915 is very similar to that of the MWRzo above ~750m is explained by the fact that above this height much fewer RASS measurements are available (as in fact presented in Fig. 10), therefore the positive impact brought by the inclusion of RASS measurements is greatly reduced above that height.

Changed.

Added.

Deleted "that".

Changed.

We think it is important to refer to the 1.5 km height because this is the maximum height reached by most of the RASS measurements.

Changed.

There is no contradiction in these lines. We use a separation level 1.5 km to highlight the different behavior of the scores: all profiles are more smoothed and uniform above 1.5 km (with MWRzo449 having the best RMSE and BIAS) but less so closer to the surface.

Section 4.2, line 570 : Please change « NN retrievals are very variable » into « the accuracy of NN

retrievals is very variable ».

Changed.

Section 4.2, line 571 : Your conclusion is only true above 1 km altitude, below 1 km altitude, NN

retrievals perform better than MWRz and MWRzo and even the two configurations with RASS

measurements. The degradation of NN retrievals above 1 km is mainly due to a large bias which might be due to the fact that you do not apply the bias correction to MWR measurements for NNs whereas you apply it to the PRs. This needs to be justified and clearly stated here. Linked to my previous comment, I do not understand how NN retrievals can be improved below 1 km with oblique measurements whereas you concluded in section 4.1 that oblique measurements present a large bias. Additionally, the MWRz using only zenith measurements also present a large bias (above

K) below 1 km altitude which seems to conclude that probably opaque channels are biased both at zenith and oblique measurements. Could you also comment on the degradation of the accuracy of

MWRzo915 between ~ 200 m and 1 km ? In figure 5 you showed an example were the RASS 915

measurements were able to improve temperature retrievals of MWRz and MWRzo above 200m but averaged over all the profiles it is not the case any more. It seems to come from a bias in your retrieval that we do not observe with MWRzo 449.

Comparison to NN profiles are moved to Appendix A where the reviewer's questions have been addressed.

The degradation of MWRzo915 above 200m is also seen in Fig. 10 of the manuscript. While the availability of RASS 449 data is almost constant from 300m to 1.6 km, RASS 915 data availability faded quickly in height with its reduction from 100% availability at 300 m to almost 10% at 1km.

Fig. 5 shows the most complicated temperature profile during XPIA, it is also a very interesting case in terms of all possible active measurements' availability, from both RASS 499 and RASS 915.

Section 4.2, figure 6 : I think the vertical blue and red lines to identify a correlation of « 1 », perfect

RMSE of 0 and bias of zero are confusing for me. The figure being already crowded, I would remove these additional coloured lines for only a vertical black dashed line for panels c and f only.

Vertical lines in Fig. 6 will be changed to black in the new version.

Section 4.2, figure 7 : I am surprised that you use oblique measurements from the MWR for humidity retrievals: can you comment on the fact that this probably violates the homogeneity assumption necessary to use low elevation angles? In general, only opaque channels are used at low elevation angles and they are not sensitive to water vapor. If you used low elevation angles, did you apply a quality-control to detect inhomogenous cloudy scenes ?

Tb obliques data are used as an average from two scans, 15 and 165. We note that most of the radiosonde launches were made in periods without liquid clouds, so the oblique scans should be similar. Also, the K-band channels from the oblique scans are not used in the retrieval, thus spatial variability in water vapor is not an issue. We only use the more opaque V-band channels for the oblique scans. Therefore we believe that our calculation of humidity retrievals is valid.

Section 4.3, line 620 : what do you mean by « weighted average over the 42 vertical heights » ?

The following text is included in the text:

"The vertical resolution of the Physical Retrievals is not uniform, with more frequent levels closer to the surface. If the data from all levels are used as the simple average, the near-surface layer will be weighted more compared to the upper levels of the retrievals. To avoid this, a vertical averaging in 0-5 km profiles is performed with separate weights at each vertical level calculated by the distance between the levels."

This is a very common validation procedure over some slice of the model with uneven vertical resolution.

Section 4.3, figure 8 : Could you comment about the potential modifications to your figure if you had calculated the statistics up to 1 km or 2 km AGL instead of 5 km ?

As you do not apply a bias correction to the NN retrievals where V-band transparent channels have a strong bias I am wondering if the conclusions are not wrongly biased for the evaluation of NN

retrievals with this averaging up to 5 km. As already mentionned previously, it is not fair
to
compare two retrievals not applied on the same dataset (one with bias correction,
another one
without). At least this should be again commented when discussing the results of this
section.

We made the statistical evaluation of temperature profiles up to 1, 2 and 5 km heights (see Fig. R9).

[Figure]

**Fig. R9**. On each double-panel plot from top to bottom: biases (retrievals minus ATkernel radiosonde), RMSEs, standard deviations of the difference between retrievals and ATkernel radiosonde, and Pearson correlations for the six PR configurations and three NN retrievals, oblique, zenith and their combination, averaged from the surface to 5 km AGL (top), to 2 km AGL (bottom left) and to 1 km AGL (bottom right), and averaged over the 15 events furthest from the priors (hatched boxes).

Statistical analysis shows similar behavior for the PR configurations in terms of RMSE for all three vertical layers. For NN statistics, we included a third type of comparison against the radiosonde measurements, the combination of the oblique scan temperature profiles up to 1 km AGL and the zenith scan temperature profiles above 1km AGL. This combined NN has the lowest RMSE compared to the other two NN scans considered separately. Also, these combined NN profiles have the lowest RMSE in the lower layer of 0-1 km compared to all PR profiles, but larger RMSE in wider atmospheric layers such as 0-2 or 0-5 km. All three NN retrievals (oblique only, zenith only, oblique and zenith combined) have the highest RMSE compared to all PR configurations in the layer of the atmosphere up to 5 km. From the PR temperature profiles, the RMSE decreases from the passive instrument configurations (MWRz, MWRzo) to the configurations with active RASS measurements in very similar ways over the 0-5, 0-2, and 0-1 km atmospheric layers, especially when comparing the 0-1 and 0-5 km layers of the atmosphere. Bias also improves from MWRz/MWRzo to the configurations that include RASS. The setting of MWRz2sigma449 shows the best statistics in terms of bias and RMSE compared to all other PR retrievals, and better to all three NN retrievals in 0-2 and 0-5 km layers. In general, almost all PR profiles with RASS have RMSE below 1 K in all three vertical layers.

Conclusion line 703 : I honestly did not have understand that NN retrievals with oblique measurements do not use the zenith observation. This has to be more explicit directly in section 2.1,
line189 to 201. This explanation arrives too late in the manuscript
Comparison to NN profiles are moved to Appendix A, where we clarified the difference between the NN configurations.

Conclusion : line 718 when MonoRTM is mentioned is redundant with line 719. I suggest modifying lines 717 to 719 into :
the small systematic errors that exist between the MWR observed Tb values and the RASS
measurements and (b) the systematic errors that exist in forward microwave radiative models.
(I would thus remove all the text between parentheses).
Modified.

Conclusion, line 722 : please correct the most difficult to retrieve and the most important to
forecast.
Corrected.

Conclusion, line 728: this sentence should be mitigated : the study proves that active sensors can
improve MWR passive observations with zenith observations only but due to the weird results you
obtain with lower elevation angles which are expected to improve the retrievals in the same area as
the RASS measurements I think you should mention that the results could be different with MWRs with elevation angles usable down to 5° above the ground. In fact, with new MWR instruments
using both zenith and low elevation angles we can expect RMSE between 0.5 and 1.5 K in the first
km (1.5 K for cloudy-scene when there is a temperature inversion in the upper layers). Thus, we
cannot be sure that the improvement brought by RASS measurements would be as much
informative in the first 1 km with a MWR unit for which oblique measurements could be optimally
used. I think you should mention this in your conclusion.
The text in the manuscript is modified as:

"Even for this subset of selected cases we find that MWRz2sigma449 produces better statistics, proving that the inclusion of active sensor observations in MWR passive observations would be beneficial for improving the accuracy of the retrieved temperature profiles also in the upper layer of the atmosphere where RASS measurements are not available (at least up to 5 km AGL). However, we note that this result may be dependent on the fact that our oblique measurements were taken at a 15 degree elevation angle, and that MWRs in locations with unobstructed views allowing for scans down to 5 degrees may provide similar improved accuracy to the temperature profiles (reference below) in 0-1 or even 0-2 km AGL layers."

Crewell, S., U. Löhnert, 2007: Accuracy of Boundary Layer Temperature Profiles Retrieved With Multifrequency Multiangle Microwave Radiometry, IEEE TGRS, VOL. 45, NO. 7, JULY 2007, **DOI:** 10.1109/TGRS.2006.888434

**Reviewer #2, answers.**

We thank the reviewer for reading our manuscript and for offering many useful comments towards its improvement. In the revised manuscript, we included modifications addressing almost all of these comments.

The submitted manuscript takes up on the ground-based remote sensing synergy approach of combining microwave radiometers (MWR) and RASS by applying a state-of-the-art physical retrieval approach. This is important, since MWR are known to show very accurate performance in temperature profiling in the lowest 500 m, whereas RASS are able to adequately capture the typical temperature inversion at the top of the atmospheric boundary layer (ABL) and thus, in theory, the synergy of both could lead to an improved temperature profile throughout the whole ABL.

**Major points**

1.) The way to showing the latter point above, however, is obviously severely hampered by the quality of the MWR data, most probably in terms of a TB bias. While the authors do show a bias correction applied to the MWR TBs, it is unclear whether this was done only for zenith observations or also at 15° elevation (Fig. 1). Here a detailed analysis is missing. If this manuscript is to be accepted for publication using real TBdata, the reason for the biases shown in Figs. 6 c and f (black a grey lines) must be identified, discussed and corrected for.

We thank the reviewer for this specific comment. Some parts of the bias-correction description were indeed missing. Additional text has been included in Section 3.2: "We compute the bias in the bias-correction procedure only from the zenith scans assuming that the same bias is suitable for the oblique scans. Also, we use the assumption that the true bias is an offset that is independent of the scene, so that the sensitivity to the scene (e.g., clear or cloudy, zenith or off-zenith) is small.  To investigate this, we eliminated the radiosondes launched during rainy periods (5 out of 58 cases) and found that the averaged temperature profiles were very little different than when all radiosonde profiles."

More detailed discussion of the temperature biases shown in Fig. 6, especially near the surface layer, will be included in Section 4.2 in the final version of the manuscript.

2.) The paper shows hardly any quantitative discussion, which is necessary for a sound scientific analysis. Except for just a few passages, discussions of the figures are carried out only in a qualitative, rather unspecific manner. With respect to this, specifically the sections 3 and 4 should be thoroughly rewritten. E.g., avoid using "This might..", "We believe…", "seemingly", "Differences", "better" or "improve" etc.

without referring to adequate statistical measures. A lot of the data is there in the XPIA data set und you can use to confirm, deny or to quantify your assumptions, respectively results.

We have tried to avoid purely qualitative descriptions, and to provide quantitative details in the indicated sections for the new version of the manuscript, thank you.

For ex., in Section 3, especially 3.2, we included the detailed descriptions of how the clear-sky days were chosen and how the uncertainty and the bias for each MWR channel were calculated.

3.) Because the authors write they could not apply any bias correction to the NN approaches, I strongly suggest omitting them from the paper. The comparisons are thus "unfair" and I do not see what benefit the reader has from including the NN retrievals when the actual goal is evaluating the MWR/RSS synergy potential that can be achieved with the PR. Instead, in all the corresponding figures, I would like to see
the results of RSS-only PR, i.e. without including the MWR so the reader has an impression what these systems are capable of in a stand-alone manner.

We thank the referee for this particular comment. Following your recommendation and also the opinion of Reviewer #1 on this matter, we decided to move all comparisons of PR and NN profiles to Appendix A, while also making note of the fact that without mentioning NN retrievals our analysis would be incomplete for the community of MWR end-users. We think that the possibility to do the bias correction in the PR is just one of the advantages the PR has. The NN retrievals are provided by the manufacturer and have the disadvantage that no bias correction is performed. They are nevertheless used by most end-users. We believe that the comparison between PR and NN is still very important and should be included in some way in the manuscript, while noting the unequal basis for the NN and bias-corrected PR comparison. These issues are now addressed in the Appendix.

Regarding a RASS-only PR, we do not see the value of this because RASS without MWR in MonoRTM will be used as RASS + prior, so we should get mostly the profile of the prior because the RASS covers only a small portion of the 17 km temperature profile. On the other hand, the RASS measurements are included in the figures, especially in Fig. 10 in the manuscript, showing what these instruments can provide in a stand-alone manner.

4.) How did you deal with clouds, what about precipitation? Did you retrieve LWP simultaneously to temperature and water vapor? What influence do clouds have on the retrieval? I find no information about this throughout the manuscript.
Most of the radiosondes were launched during clear-sky time. See also answers to Referee #1 about this.
We included several new paragraphs in Section 3.2, e.g.:
"we found that from 58 radiosonde launches used in our statistical analysis, 41 belong to the clear-sky category, 12 - to cloudy but non-precipitating conditions and 5 - to rainy periods".
A discussion of the impacts of clouds on the retrieval is mentioned in comment 1) above and will be included in the manuscript.

5.) The sections describing microwave radiometry need more background and scientific accuracy.

We had already included many references in order to avoid a detailed description of the basic principles of microwave radiometry.

**Further specific points and questions to be addressed**

1.) Abstract, last paragraph: It is not clear if the improvements described refer to the PR compared to the NN or the MWR+RASS combination compared to the MWR-only retrieval.
As we moved the discussion of PR and NN profiles comparison in Appendix A, this paragraph has been changed to highlight the purpose of this paper as:
"Having the possibility to combine the information provided by the MWR and RASS systems, in this study the physical-iterative approach is tested with different observational inputs: first using data from surface sensors and the MWR in different configurations, and then including data from the RASS. These temperature retrievals are assessed against 58 co-located radiosonde profiles. Results show that the combination of the MWR and RASS observations in the physical-iterative approach allows for a more accurate characterization of low-level temperature inversions compared to the physical retrievals of the MWR passive measurements, and that these retrieved temperature profiles match the radiosonde observations better than the temperature profiles retrieved from the MWR in the atmospheric layer between the surface and 5 km AGL.  Specifically, in this layer of the atmosphere, both root mean square errors and standard deviations of the difference between radiosonde and retrievals that combine MWR and RASS are improved by ~0.5 K compared to the difference between radiosonde and MWR retrievals. Pearson correlation coefficients are also improved.
We provide the comparison of the temperature physical retrievals to the neural network retrievals in Appendix A."

2.) Introduction: A description of the physical principle that allows temperature (& humidity) profiling (and LWP retrieval) from passive MWR observations is missing. When doing so, please consider reformulating the advantages and disadvantages of the MWR retrieval methodology, because they are currently not scientifically sound.
Be sure to differentiate how the frequency dependence and elevation angle dependence of TB can both lead to resolving the temperature profile in the vertical.
We are not sure what the Reviewer is suggesting here. There are many articles describing the MWR temperature and humidity retrievals as well as physical principles of such retrievals, and we had already included many of these references in the manuscript in order to avoid a detailed description of the basic principles of microwave radiometry.
Nevertheless, we include a description of the temperature retrieval frequencies in Section 2.1:
"V-band frequencies or channels also could be divided in two categories: the opaque channels, 56.66 GHz and higher, which are more informative in the low layer of the atmosphere from the surface to ~1 km above the ground and the transparent channels, 51-56 GHz, which are more informative above 1 km in the temperature profile".

3.) Line 109: MWR don't "apply radiative transfer equations and neural network retrievals…" – please reformulate.
This paragraph is reformulated to: "Radiative transfer equations are commonly used to train statistical retrievals or as forward models within physical retrieval methods".

4.) Line 115: Please make clear what you mean with "deep layer of the atmosphere".
Changed to: "the layer of the whole troposphere ".

5.) Section 2.1, lines 203-204: The purpose of using observations at 15° elevation is not to "average out small scale horizontal inhomogeneities of the atmosphere" but to obtain TB observations at different optical depths.
This paragraph has been modified according to your suggestion:
"In this study we make use of the NN zenith and of the NN oblique measurements, where the latter can obtain TB observations at different optical depths."

6.) Section 3.1, lines 280-286: Why does the Y vector and the error covariance matrix contain both "zenith" and "zenith+oblique" components. If I understand correctly, you can choose to use only zenith observations and add the off-zenith (=oblique) TBsto improve the retrieval? So then should it not be "zenith" and "oblique"? Please clarify.
Table 1 as well as observational vectors Y2, Y3 and Y4 and matrix Sε are modified:

| | $T_{sfc}$ | $Q_{sfc}$ | $Tb_{zenith}$ | $Tb_{oblique\_avrg}$ | $TV_{RASS915}$ | $TV_{RASS449}$ |
|---|---|---|---|---|---|---|
| $Y_1$ = MWRz | X | X | X | | | |
| $Y_2$ = MWRzo | X | X | X | X | | |
| $Y_3$ = MWRzo915 | X | X | X | X | X | |
| $Y_4$ = MWRzo449 | X | X | X | X | | X |

$$Y_1 = \begin{bmatrix} T_{sfc} \\ Q_{sfc} \\ Tb_{zenith} \end{bmatrix} \qquad Y_2 = \begin{bmatrix} T_{sfc} \\ Q_{sfc} \\ Tb_{zenith+oblique\ avrg} \end{bmatrix}$$

$$Y_3 = \begin{bmatrix} T_{sfc} \\ Q_{sfc} \\ Tb_{zenith+oblique\ avrg} \\ Tv_{RASS915} \end{bmatrix} \qquad Y_4 = \begin{bmatrix} T_{sfc} \\ Q_{sfc} \\ Tb_{zenith+oblique\ avrg} \\ Tv_{RASS449} \end{bmatrix}$$

$$S_\varepsilon = \begin{bmatrix} \sigma^2_{Tsfc} & 0 & 0 & 0 \\ 0 & \sigma^2_{Qsfc} & 0 & 0 \\ 0 & 0 & \sigma^2_{Tb_{zenith}}\ \textcircled{1}\ or\ \sigma^2_{Tb_{zenith+oblique\ avrg}}\ \textcircled{2} & 0 \\ 0 & 0 & 0 & \sigma^2_{Tv_{RASS915(449)}}\ \boxed{3\ or\ 4} \end{bmatrix}$$

7.) Line 310: Do you mean the covariance between the uncertainties of the measurements?

This part of the manuscript is reformulated:

"The uncertainty in the MWR Tb observations was set to the standard deviation from a detrended time-series analysis for each channel during cloud-free periods.  The derived uncertainties ranged from 0.3 to 0.5 K in the 22 to 30 GHz channels, and 0.5 to 1.0 K in the 52 to 60 GHz channels. We assumed that there was no correlated error between the different MWR channels.

For the RASS, collocated RASS and radiosonde profiles were compared and the standard deviation of the differences in Tv were determined as a function of the radar's signal-to-noise ratio (SNR). This relationship resulted in uncertainties that ranged from 0.8 K at high SNR values to 1.5K at low SNR values. Again, we assumed that there was no correlated error between different RASS heights. Following all these assumptions, the covariance matrix $S_\varepsilon$ is diagonal."

8.) Section 3.2: There seems to be a non-consistent use of terminology. Please use "uncertainty" only in the sense of random uncertainty and distinguish it clearly from systematic offset (=bias).

We have made certain to consistently refer to the random uncertainty of Tb as the uncertainty, and the systematic offset as the bias.

9.) Lines 323-324: erroneous, please reformulate in a consistent manner

The text is changed as:

"While the bias of the retrieval depends on both the sensitivity of the forward model and the observational systematic offset, we can try to eliminate, or at least to reduce, the systematic error in the MWR observations."

10.) Line 327: The 30 GHz channel is not predominantly water vapor, but liquid water sensitive.
Changed.

11.) Lines 328-330: "The random uncertainty in brightness temperature was calculated as its standard deviation during clear sky times and for this channel is approximately 0.3 K": Why is this calculated standard deviation related to the TB uncertainty? Over what time window did you average? What about water variability in the atmosphere during the calculation time? Why actually did you calculate this standard deviation and where do you use it in the course of your study?
Thank you for this comment. We included a much more detailed description of the uncertainty calculation in the text in Section 3.2:
"A threshold value of 0.3 K has been used for the uncertainty calculation. The random uncertainty in Tb is calculated as an average of the Tb standard deviation in a one hour sliding window through all data points of a day. It also could be computed as the standard deviation of the difference between Tb and the smoothed Tb to eliminate daily temperature variability. Finally, there is a "standard" set of uncertainties used as the high boundaries for Tb uncertainty per MWR channels calculated empirically in the previous experiments. Four clear-sky days have been chosen using a criterion of 0.3 K uncertainty in the 30 GHz channel: March 10 and 30, and April 13 and 29, 2015.
During periods with liquid-bearing clouds overhead, this criterion is markedly higher (more than 0.7 K) and much higher for the rainy periods (> 4 K). While those calculations were applied on a daily basis, it is important to mention that the days are not uniform in terms of cloudiness or rain.  Therefore, we used the data for the 2-3 hours bracketing the time of radiosonde launches to determine to which category a particular radiosonde profile belongs, clear-sky, cloudy or rain.  In this way, we found that from 58 radiosonde launches used in our statistical analysis, 41 belong to the clear-sky category, 12 - to cloudy but non-precipitating conditions and 5 - to rainy periods. For the four chosen clear-sky days not only were the daily uncertainties of 30 GHz Tb below 0.3 K, but all three sets of uncertainties described above were extremely similar with the averaged difference less than 0.05 K."

12.) Lines 332-333: How were the clear-sky days selected?
Please, see above.

13.) Lines 333-334: How did you calculate the bias?
From the modified text in Section 3.2:
"The bias was computed for each of the 22 channels as the averaged difference between the observed Tb from the MWR zenith observations, and the forward model calculation applied to the prior, over these selected clear-sky days, and then subsequently removed from all of the MWR observations."

14.) Before line 358: a description and a quantitative discussion of the Sa and Se matrices applied needs to be given before going on describing retrieval results.

Sa and Se matrices are described in Section 3.1 and retrieval results are discussed in 3.2.

15.) Lines 425 and following, referring to Fig. 3: quantitative argumentation missing and VRES "jumps" in Fig. 3 are not discussed

We thank the Reviewer for this comment very much because it prompted us to reconsider the method used to calculate the vertical resolution.

There are two ways to compute the vertical resolution from the averaging kernel. First, we applied a method that Tim Hewison published (TGRS 2007, reference below) that uses only the diagonal data of the averaged kernel. It works well when the retrieval uses only the input from the passive observations, like the MWR, but is not very suitable for the passive/active combination of inputs, as is seen in Fig. 3d in the manuscript (with the creation of the "jumps"). So, we returned to the method (that we actually erroneously mentioned in the paper) that computes the vertical resolution as the full-width half-maximum (FWHM, Maddy and Barnet, TGRS, 2008, reference below) value of the averaging kernel at each height.

T. J. Hewison, "1D-VAR Retrieval of Temperature and Humidity Profiles From a Ground-Based Microwave Radiometer," in IEEE Transactions on Geoscience and Remote Sensing, vol. 45, no. 7, pp. 2163-2168, July 2007, doi: 10.1109/TGRS.2007.898091.

Maddy, E. S. and C. D. Barnet, 2008: Vertical Resolution Estimates in Version 5 of AIRS Operational Retrievals. IEEE TGRS, VOL. **46**, NO. 8, AUGUST 2008, doi:10.1109/TGRS.2008.917498

Using the FWHM method, Fig.3 is changed to the one below, where the "jumps" in panel d are significantly reduced:

[Figure]

16.) Section 4.1, lines 469-471: unspecific sentence, please reformulate

This sentence is deleted because soon after the similar text is followed:

"MWRzo449 has the best statistical measures compared to the other PRs, particularly below 2 km AGL, where RASS 449 measurements are available".

17.) Fig. 5: How many cases are used for the statistics, how many are clear-sky, how many are cloudy sky? How did you deal with cloudy cases in general?

Statistical results are shown in Figs. 4, and 6-10, not in Fig. 5 of the manuscript (where a single case profile - 18 March, 2015 at 0200UTC is presented). For the statistical analysis, from 58 valid radiosonde profiles 41 have been launched in clear-sky periods, 12 - in cloudy but non-precipitating conditions and 5 - in rainy time. This information is now included in the manuscript, Section 3.2. We defined those categories using the 30 GHz channel Tv as in these figures:

[Figure]

Zenith Tb from a 30 GHz channel for a clear-sky day (left panel), cloudy day (middle panel) and rainy day (right panel) from the CU radiometer in red and NOAA radiometer in blue. STDDEV(Tb-SMOOTH(tb,11)) is shown at the bottom in each panel with its average values printed under the panels in corresponding colors. Vertical (green – for clear-sky, beige – for clouds and cyan – for rain) lines show the time of radiosonde launches.

18.) Fig. 8: Can you derive meaningful statistical measures such as RMSE from only 15 cases?

This is a valid comment. We are interested in describing the "worst case" most extreme events, when the radiosonde temperature profiles are most different from the prior profile, and so, by definition the number of cases needs to be limited, otherwise they are no longer extreme. On the other hand, some level of statistical significance is desired. Given that we have 58 radiosondes, 15 events are already nearly 25% of the total. We felt that this was a reasonable compromise given the limitations of the data set.

19.) Fig. 9: The MWRz2sigma449 performs best compared to the other retrievals. This retrieval relies on an increase in the MWR uncertainty, which was chosen in an arbitrary manner. This choice should be thoroughly justified and set into context with the performance of the 449-only retrievals which I would like to see (see "Major points" above).

The choice of double MWR uncertainty for MWRz2sigma449 is not arbitrary, but the reviewer is absolutely right, it is not qualitatively justified in the manuscript. It was chosen based on the "worst" XPIA temperature profile on March 18, 2015, 02:00 UTC showing in Fig.5

in the manuscript. This particular case is not only the worst in the XPIA experiment in terms of temperature inversions (three of them in one profile, with one near the surface), but with other complications. We found that the MWR Tb from the opaque channels of both zenith and obliques scans, have biases (to the forward model calculation of radiosonde Tb) of around 1 K. We wanted to check our hypothesis about too little freedom of the PR approach in the layer between surface and RASS measurements. As is mentioned in the text, "After several trials", we indeed made many additional runs, but we wanted to keep our recommendations general, and not be very specific about this particular case.

20.) Section 4.4, lines 683-686: This sentence is formulated in a general, rather nonspecific way and could be given without any of the studies conducted here.
This paragraph is removed.

**Technical comments**

1.) Figures are given in rather low resolution, a higher one would have been nice to be able to better interpret the results.
All figures are in tiff format that has a high resolution. The deterioration of the images comes from the conversion to PDF. Original tiff format files will be provided to the editorial office when requested.
2.) Equation fonts appear in a non-standard, unorganized way.
Equation font is changed to be the same throughout the paper.

3.) In general: please write K or °C, but not °K.
Checked and fixed.

4.) Section 3.1, lines 280-286: Numerate all equations, be consistent with equation fonts and text fonts, be consistent with variables (i.e. L, LWP), explain all variables (and indices) in the text. Please be neater.
Lines 280-286 consist of only one equation, Eq. (1), which is numbered, and the descriptions of all its terms. We changed the text to have consistency in fonts and text fonts, and we consistently used LWP in the revised text.

5.) Line 348 and following: use a new sub-section, the paragraphs are not related to "Bias-correction" anymore
We renamed the Section **3.2 PR's bias-correction** to **3.2 PR's bias-correction and PR's temperature profiles.**

6.) Section 3.3, lines 415-421: move text to Fig. 3 caption
The text on these lines reformulates the Fig. 3 description in a more explanatory way.

[revised manuscript text omitted]

---

## Referee Report (RR1)

**Improving thermodynamic profile retrievals from microwave radiometers by including Radio Acoustic Sounding System (RASS) Observations**

**General Comment :**

The authors have replied to most of my comments and most questionable aspects . The new version where the discussion on NN retrievals is moved to a dedicated Appendix seems to be better suited due to the lack of bias-correction for the NN retrievals but still interesting results to show. I still recommend to answer a few questions about the MWR data bias correction. As biases of the data used in this study are quite large, I think it is important to clearly discuss and clarify the choices made by the author.

After answering these few points, I would recommend the publication of the manuscrit to AMT.

Most of the figures are also of poor quality and difficult to read in the pdf version. The authors mention that it is only due to the pdf formatting but original figures are of good quality. I let the editor checks this point in the final version.

1) I am still a bit puzzled by the bias-correction calculation. This is an important aspect of the manuscript as the authors mention the large bias affecting MWR opaque channels of the MWR_NOAA unit that could make sub-optimal the analysis presented in this paper.
If I understand properly the bias has been computed from differences with the « prior » profiles and thus a climatological montly mean. Biases in the most opaque channels are significantly affected by the accuracy of the boundary layer temperature profiles used in the simulation by night when low level inversions are present. I am not convinced that a monthly average could correctly infer the bias in the most opaque channels if such cases (low level temperature inversions) are taken into account in the statistics. Similarly K-band and transparent V-band channels are sensitivity to the integrated water vapour which might be poorly represented by a monthly average too.
   - Could you explain why clear-sky radiosondes profiles of the campaign cloud not be used to infer this bias correction ?
   -  Did you investigate the sensitivity of the bias-correction to the database used to derive this correction (prior profiles versus radiosondes) ?
   - From figure R2 of your answer : are you convinced that the bias is the same at zenith and oblique scans for this case study (I am sorry but this is very hard for me to read the numbers in the y-axis to check if the values look approximately the same at zenith and oblique scans) ?
   - Did you check that the calculated biases were approximately the same through the whole period at zenith and oblique scans as you mention line 420 ?
   - Figure5 : in your answer, you show that the oblique scan at 165° is probably affected by a cloud that is probably not yet detected at 15°. Could not that be a problem in the retrieval to mix two TB measurements : one-clear-sky and one cloudy-sky ? How does the PR handle this ? I think resolving elevated temperature inversions often observed during stratus cloud is already challenging for MWRs but probably even more if you mix two scans one in clear-sky and one in cloudy-sky.
   - Line 601 : you mention that the bias appears « in this case » : can you just clarify that it is probably affecting the whole time series and not only the case shown in figure2

In figure R3 of your answer you clearly demonstrate that the problem of the MWR retrievals comes from the large bias in the MWR_CU unit that would be mostly solved with the MWR_NOAA unit. I would remove channel 57.2884 GHz from the analysis as this channel is likely affected by an

hardware problem. But I think including this result in your manuscript would be very interesting to clearly demonstrate your hypothesis for the reader (like it is clearly demonstrated in your reply)

2) line 318 : The uncertainty in MWR observations is evaluated as the standard deviation of Tb measurements during clear-sky measurements. Firstly could you specify the time window for this uncertainty calculation and secondly could you discuss about forward model uncertainties ?
Are these taken into account in the observation error covariance matrix ? I

3) Line 410 : you mention that 5 RS are under rainy conditions. Could you confirm that these data have been discarded from the results in figures 6 to 9 (statistical analysis) ?

---

## Referee Report (RR2)

**Review AMT-2021-9:**
**Improving thermodynamic profile retrievals from microwave radiometers by including Radio Acoustic Sounding System (RASS) Observations**

First of all I would like to thank and congratulate all the authors of the manuscript for taking into account the major comments of the previous versions and for providing such an improved version. I appreciate all their efforts to improve the equations, figures, scientific evaluation and discussion as well as new results proposed with the MWR bias correction impact.
I only suggest a few minor corrections for typos and I think probably a mistake in the text.
I recommend the publication of the manuscrit to AMT.

**Minor corrections :**

**Line 337** : we have demonstrated the augmentation → we have demonstrated the extension

**Line 447** : It is mentionned that the BC is computed with the TROPOe retrievals. I doubt this would be a good idea as the TROPOe retrieval is obtained to minimize the distance with the observation. So, if the observations is biased, we of course end with an atmospheric profile compensating the bias in the observation. Thus, I assume that the difference between the observation and the simulation from the TROPOe retrievals should tend to zero as the PR want to minimize this distance. I think the authors wanted to mention TOPROe background profiles from the climatology as it is confirmed in line 498 and explained in their reviewer's answers. Please correct the text accordingly.

**Line 635** : as a function of the height → as a function of height

**Figure 8** : bottom right : change 0-5 km averaged into 0 – 3 km averaged.

---

## Author Response (AR2)

**Review AMT-2021-9:**
**Improving thermodynamic profile retrievals from microwave radiometers by including Radio Acoustic Sounding System (RASS) Observations**

**ANSWERS to REFEREE 1**

**General Comment:**
The authors have replied to most of my comments and most questionable aspects. The new version where the discussion on NN retrievals is moved to a dedicated Appendix seems to be better suited due to the lack of bias-correction for the NN retrievals but still interesting results to show. I still recommend to answer a few questions about the MWR data bias correction. As biases of the data used in this study are quite large, I think it is important to clearly discuss and clarify the choices made by the author.
After answering these few points, I would recommend the publication of the manuscript to AMT.

We wish to thank very much both Referees for carefully reading our manuscript and for offering many comments towards its improvement. In revising the manuscript, we have considered all of their comments. Many changes have been applied to the current version of the manuscript and we believe that our manuscript benefited significantly from the constructive comments made by both the Referees. Please, find below our point-to-point answers in red.

Most of the figures are also of poor quality and difficult to read in the pdf version. The authors mention that it is only due to the pdf formatting but original figures are of good quality. I let the editor check this point in the final version.

In the revised version of the manuscript we have made sure to present figures clearly, both in the format and in the quantitative discussion of the results. Same for the equations. Nevertheless, in the final version of the manuscript we'll make sure to provide the Editorial office with the high-resolution version of the images in *.eps format and Equations from Microsoft's equation editor.

1) I am still a bit puzzled by the bias-correction calculation. This is an important aspect of the manuscript as the authors mention the large bias affecting MWR opaque channels of the MWR_NOAA unit that could make sub-optimal the analysis presented in this paper.
If I understand properly the bias has been computed from differences with the « prior » profiles and thus a climatological monthly mean. Biases in the most opaque channels are significantly affected by the accuracy of the boundary layer temperature profiles used in the simulation by night when low level inversions are present. I am not convinced that a monthly average could correctly infer the bias in the most opaque channels if such cases (low level temperature inversions) are taken into account in the statistics. Similarly, K-band and transparent V-band channels are sensitivity to the integrated water vapor which might be poorly represented by a monthly average too.

We have taken this comment from the Referee very seriously and have modified large parts of the manuscript to address these concerns. The Referee is correct that a dataset like XPIA has the advantages to have many radiosondes that could be used instead of the prior to reduce any observational systematic offsets in the MWR Tbs. While this is true for the XPIA dataset, for other field campaigns this approach might not be possible because radiosonde observations are not always available. For this reason, the Section regarding the bias correction of the MWR measured Tbs has been completely rewritten and renamed *"3.2 Bias-correction of MWR observations using radiosondes or climatology"*. In this section, two bias-correction methods are now presented, the first using radiosonde data (radiosonde BC) and the second using the prior (TROPoe BC). Through the revised manuscript, results using the two different bias-correction procedures are presented and analyzed. As expected, we find that the radiosonde BC method gives retrieved profiles closer to the radiosonde temperature profile than when using TROPoe BC.

Although the more accurate radiosonde BC method has been included in the revised version of the manuscript, the final goal of this study is not to assess the sensitivity to different bias-correction approaches, but to verify that the inclusion of RASS observations does improve retrieved temperature profiles. We show that this result is still valid independently of the bias-correction method used.

•Could you explain why clear-sky radiosondes profiles of the campaign could not be used to infer this bias correction?

Following the Referee's comments, as mentioned in the answer above, in the revised version of the manuscript two bias-correction methods are now presented, the first using radiosonde data (radiosonde BC) and the second using the prior (TROPoe BC). When using the radiosonde BC method, the procedure to identify the clear-sky days is now one-hour centered around the radiosonde times. On the other hand, when using the TROPoe BC, to identify clear-sky days we keep using the same procedure as before, to consider a field campaign scenario that does not benefit from so many radiosonde launches as the XPIA campaign.

• Did you investigate the sensitivity of the bias-correction to the database used to derive this correction (prior profiles versus radiosondes)?

Following the Referee's comments, as mentioned in the answers above, we did indeed investigate the sensitivity of the bias-correction to the database used to derive this correction (radiosonde BC versus TROPoe BC), including the results obtained from both approaches in the revised version of the manuscript. A direct comparison of the different biases is shown in Fig. 1 of the revised manuscript.

•From figure R2 of your answer: are you convinced that the bias is the same at zenith and oblique scans for this case study (I am sorry but this is very hard for me to read the numbers in the y-axis to check if the values look approximately the same at zenith and oblique scans)?

Following this Referee's comments and also the comments from the other Referee, we did measure the bias in the MWR observed Tbs (in all 22 MWR channels of the zenith scan, and in the four opaque channels of the oblique scans, i.e. the channels used by TROPoe), in comparison to radiosonde Tbs (radiosonde BC) and in comparison to the TROPoe Tbs (TROPoe BC). These results are now presented in Fig. 1 of the revised manuscript, which we hope it is easier to read. The following description of Fig. 1 has been included in the text: *"The biases from the two bias-correction schemes are within the uncertainties of each other for most of the channels except at the higher frequencies in the V-band. Biases in the most opaque channels are significantly affected by the accuracy of the boundary layer temperature profiles. When TROPoe BC is used, a monthly average prior temperature profile is used in the PR, and thus differences between this prior profile and the actual temperature profile can result in a spectral bias in the more opaque MWR channels. On the contrary, the radiosonde BC uses a direct measurement of the temperature profile (from the radiosonde), and thus is more accurate. It is also important to note that, in both approaches, the biases in the opaque channels for zenith and for oblique scans (for radiosonde BC these are red and blue, respectively; and for the TROPoe BC these are black and green, respectively) are very similar to each other. **This supports the assumption that the true bias is nearly independent of the scene, or that the sensitivity to the scene (e.g., zenith or off-zenith) is small."***

•Did you check that the calculated biases were approximately the same through the whole period at zenith and oblique scans as you mention line 420?

As shown in Fig.1 of the revised manuscript, the standard deviation over all clear-sky radiosonde profiles (red error bars for the zenith scan) are around 0.1-0.2 K in the opaque channels, which demonstrate that the biases do not change much during the duration of the XPIA campaign.

•Figure5: In your answer, you show that the oblique scan at 165° is probably affected by a cloud that is probably not yet detected at 15°. Could not that be a problem in the retrieval to mix two TB measurements: one-clear-sky and one cloudy-sky? How does the PR handle this? I think resolving elevated temperature inversions often observed during stratus clouds is already challenging for MWRs but probably even more if you mix two scans one in clear-sky and one in cloudy-sky.

This is a very good point. In the revised version of the manuscript we identify the clear-sky period looking at the zenith scan and the averaged oblique scans separately. Data from the zenith scan indicates more than 35 radiosonde profiles in clear-sky conditions, but the same evaluation of the oblique scans reduces this number to 18. We decided to use only these 18 radiosonde profiles so that we are sure that all scans (15°, 90°, 165°) observe clear-sky conditions.

•Line 601: you mention that the bias appears « in this case »: can you just clarify that it is probably affecting the whole time series and not only the case shown in figure2

According to the data in Fig. 1, the biases from the two bias-correction schemes are within the uncertainties of each other for most of the channels except at the higher frequencies in the V-band, where there are almost 1°C differences between the TROPoe biases and radiosonde biases for four opaque channels. The radiosonde data are the direct measurements, so as TROPoe Tb biases do not match radiosonde biases for opaque channels, we may confirm that TROPoe biases couldn't correct Tb data perfectly and therefore the cases like one shown in Fig. 5, are not unique but could affect the whole data set.

1) In figure R3 of your answer you clearly demonstrate that the problem of the MWR retrievals comes from the large bias in the MWR_CU unit that would be mostly solved with the MWR_NOAA unit. I would remove channel 57.2884 GHz from the analysis as this channel is likely affected by a hardware problem. But I think including this result in your manuscript would be very interesting to clearly demonstrate your hypothesis for the reader (like it is clearly demonstrated in your reply)

We decided to keep using the MWR CU units in the study only because the other unit was unavailable for almost the entire month of April. Nevertheless, we do believe that following the Referees' comments (i.e., including the radiosonde BC method to the study) largely demonstrated that these biases could be corrected (see Fig. 5a of the revised manuscript), and the assumed hypothesis, that the inclusion of the RASS observations to the retrievals improves the temperature profiles, could still be verified.

2) line 318: The uncertainty in MWR observations is evaluated as the standard deviation of Tb measurements during clear-sky measurements. Firstly, could you specify the time window for this uncertainty calculation and secondly could you discuss about forward model uncertainties? Are these taken into account in the observation error covariance matrix?

The uncertainties in MWR observations are calculated differently for two bias-correction methods but both methods use the time-series data in the 30 GHz channel to identify the clear-sky time period. For radiosonde BC, the 18 launch times are chosen based on the small (<0.4°C) standard deviation of Tb in one-hour data centered at the radiosonde launch time, from both scans, zenith and oblique. TROPoe BC uses several clear-sky days (four in our case). The uncertainty in the MWR Tb observations was set to the standard deviation from a detrended time-series analysis for each channel during these cloud-free periods.
Regarding the forward model uncertainties, we do mention in the manuscript that *"The bias of the retrieval depends on both the absolute accuracy of the forward model and on any observational systematic offset"*. Additionally, since in the revised version of the manuscript we are now presenting the posterior covariance matrix, Sop, we use it to provide a measure of the uncertainty of the retrievals (Figs. 3 and 4, especially Fig. 4e).

3) Line 410: you mention that 5 RS are under rainy conditions. Could you confirm that these data have been discarded from the results in figures 6 to 9 (statistical analysis)?

Yes, we do confirm that the rainy conditions have been excluded from the results presented in the statistical analysis. This has now been clarified in the text: "*Moreover, while accurate RASS*

*data can be collected during rain, MWR data could be potentially deteriorated due to water deposition on the radome. Therefore, six profiles (three for March 13, and one each on May 1, 3 and 4) were eliminated from the statistical evaluation. These restrictions lowered the number of total radiosonde launches used in this study to 52"*

**Review AMT-2021-9:**
**Improving thermodynamic profile retrievals from microwave radiometers by including Radio Acoustic Sounding System (RASS) Observations**

**ANSWERS to REFEREE 2**

This submission has great potential, but from my point it cannot be published like this. There are just too many basic things that still don't live up to a peer-reviewed scientific publication, starting from all sorts of formal issues over unclear & erroneous methods to non-scientific ways of argumentation.

We wish to thank very much both Referees for carefully reading our manuscript and for offering many comments towards its improvement. In revising the manuscript, we have considered all of their comments. Many changes have been applied to the current version of the manuscript and we believe that our manuscript benefited significantly from the constructive comments made by both the Referees. Please, find below our point-to-point answers in red.

General: Formal issues, equations, figures and figure descriptions don't fulfill the general standards; are there is still not enough quantitative discussion.

In the revised version of the manuscript we have made sure to present figures clearly, both in the format and in the quantitative discussion of the results. Same for the equations. Nevertheless, in the final version of the manuscript we'll make sure to provide the Editorial office with the high-resolution version of the images in *.eps format and Equations from Microsoft's equation editor.

Below please find my general major points, without going into the details of the Result section.

We have taken the Referees' comments under consideration and have modified the manuscript accordingly. The Results Section has also been modified as a consequence of the changes applied throughout the whole manuscript and we do believe that the flow of the manuscript, as well as the presentation of the results, is much improved as a consequence of these changes.

Lines 198-200: Still no reason (or reference) given why MWR off-zenith TBs improve the temperature profile, respectively are used.

Crewell and Löhnert (2007) demonstrated conclusively that the addition of elevation scans improves the accuracy of the retrieved temperature profiles. Furthermore, this improvement is because the elevation scans increase the information content in the retrievals, and is shown in other papers (e.g., Turner and Löhnert 2021). Most MWR vendors and groups routinely use elevation scans in their temperature retrievals as part of their usual operating procedure.

Section 3.1: (too) many formal issues

Section 3.1 and the equations wherein have been checked, hopefully fixing the "formal issues" mentioned by the Referee.

Line 384: Why are "Physical retrieval bias-correction" and "temperature profiles" included in one sub-section?

We do agree with the Referee that the previous organization of Sections 3 and 4 (and their Subsections) was not straightforward, so we did reorganize the manuscript completely in the revised version. We changed it to:

3. Physical retrievals
    3.1    Iterative retrieval technique
    3.2    Bias-correction of MWR observations using radiosondes or climatology
    3.3    Analysis of physical retrieval characteristics
4. Results
    4.1    Statistical analysis of the physical retrievals up to 3 km AGL
    4.2    Statistics for the profiles least close to the climatology
    4.3    Virtual temperature statistics

With this new organization we do believe that the flow of the manuscript should be improved.

Lines 384-389: The terms bias and uncertainty seem mixed up.

Fig. 1 has been modified in the revised manuscript, as well as the text referring to it: "*Fig. 1 shows the Tb biases found for all 22 MWR channels from both bias-correction approaches. The biases calculated with the radiosonde BC scheme are shown for all channels used in our analysis: 22 channels of the zenith scan, in red, and four V-band opaque channels of the oblique scans, in blue. The black and green triangles represent the biases calculated using the TROPoe BC approach for zenith and for zenith+oblique scans, respectively. **All biases are presented with associated uncertainties**.*"

Line 351: "bias of the retrieval depends on both the sensitivity of the forward model…" Do you mean the absolute accuracy of the forward model or really the sensitivity? If the latter: sensitivity to what?

**Commented [1]:** Text on lines 384-389 was: "Fig.1. Bias for the four chosen clear-sky days (red-dashed lines) and their mean (red solid line) for the original observations in the top panel, and for the bias-corrected data in the bottom panel. Green lines are the uncertainty boundaries around the mean bias. Frequencies used in the PR algorithm are marked with black triangles in both panels."

*The Referee is correct and we have modified the mentioned sentence to: "The bias of the retrieval depends on both the absolute accuracy of the forward model and on any observational systematic offset".*

Line 351-370: The method described in this paragraph is scientifically not sound: I am puzzled about mixing up the terms systematic error reduction, uncertainty determination and clear & cloudy-sky determination.

*We have taken the comments of both Referees very seriously on these points and have modified large parts of the manuscript to address their concerns. For this reason, the Section regarding the bias correction of the MWR measured Tbs has been completely rewritten and renamed "3.2 Bias-correction of MWR observations using radiosondes or climatology". In this section, two bias-correction methods are now presented, the first using radiosonde data (radiosonde BC) and the second using the prior (TROPoe BC). In this updated Section we have now explained in detail what are the steps in each of the bias-correction approaches, starting with the clear-sky days identification, moving to the MWR Tbs biases calculation and the bias removal.*

Lined 371-383: A bias calculation should not be carried out with respect to the prior. Also: need to show with measurements that bias in zenith and off-zenith TBs do not differ, a mere assumption here is not enough.

*Following both Referees' comments, and as mentioned in the answer above, in the revised version of the manuscript two bias-correction methods are now presented, the first using radiosonde data (radiosonde BC) and the second using the prior (TROPoe BC). Additionally, we did determine the bias in the MWR observed Tbs (in all 22 MWR channels of the zenith scan, and in the four opaque channels of the oblique scans, i.e. the channels used by TROPoe), in comparison to radiosonde Tbs (radiosonde BC) and in comparison to the TROPoe Tbs (TROPoe BC). These results are now presented in Fig. 1 of the revised manuscript. The following description of Fig. 1 has been included in the text: "The biases from the two bias-correction schemes are within the standard deviation of each other for most of the channels except at higher frequencies. [...] It is also important to note that, in both approaches, the similarity between the biases in the opaque channels for zenith and for oblique scans [...] are very similar to each other.* **This supports the assumption that the true bias is nearly independent of the scene, or that the sensitivity to the scene (e.g., zenith or off-zenith) is small."**

Lines 428-431: The averaging kernel must be applied to the radiosonde profiles, not to the retrievals

*The Referee is of course right, and we corrected the error accordingly.*

Fig. 2: not explained how why and how smoothed radiosonde profiles differ, this would involve

**Commented [2]:** Text on lines 351-370 was: "While the bias of the retrieval depends on both the sensitivity of the forward model and the observational systematic offset, we can try to eliminate, or at least to reduce, the systematic error in the MWR observations. To this aim, we first looked for clear sky days (to reduce the degrees of freedom associated with clouds) during the period of the measurements. One method to identify clear-sky times is to use Tb observations in the 30 GHz liquid water sensitive channel. The random uncertainty in Tb is calculated as an average of the Tb standard deviation in a one-hour sliding window through all data points of a day. (It also could be computed as the standard deviation of the difference between Tb and the smoothed Tb to eliminate daily temperature variability.) Four clear-sky days have been chosen using a criterion of 0.3 K uncertainty in the 30 GHz channel: March 10 and 30, and April 13 and 29, 2015. During periods with liquid-bearing clouds overhead, this criterion is markedly higher (more than 0.7 K) and much higher for the rainy periods (> 4 K). While those calculations were applied on a daily basis, it is important to mention that the days are not uniform in terms of cloudiness or rain. Therefore, we used the data for 2-3 hours around the time of radiosonde launches to determine to which category a particular radiosonde profile belongs, clear-sky, cloudy or rain. In this way, we found that from 58 radiosonde launches used in our statistical analysis, 41 belong to the clear-sky category, 12 - to cloudy but non-precipitating conditions, and 5 - to rainy periods. For the four chosen clear-sky days not only were the daily uncertainties of 30 GHz Tb below 0.3 K, but both sets of uncertainties described above were extremely similar with the averaged difference less than 0.05 K.

**Commented [3]:** Text on line 371-383 was: "The bias was computed for each of the 22 channels as the averaged difference between the observed Tb from the MWR zenith observations, and the forward model calculation applied to the prior, over these selected clear-sky days, and then subsequently removed from all of the observations. We compute the bias in the bias-correction procedure only from the zenith scans, assuming that the same bias is suitable for other scans. Also, we assume that the true bias is an offset that is nearly independent of the scene, so that the sensitivity to the scene (e.g., clear or cloudy, zenith or off-zenith) is small. To investigate that, we eliminated the radiosondes launched during rainy periods (5 out of 58 cases) and found that the average temperature profiles were very little different than when all radiosonde profiles were used, with the maximum bias and RMSE absolute differences 0.12 K and 0.11 K respectively up to 5 km AGL. Fig. 1 shows the results of the bias-correction for the four chosen clear-sky days. The green lines on this figure indicate the MWR random errors; these are 0.3-0.4 K for K-band channels and 0.4-0.7 K for V-band channels."

explaining how the Jacobians are built up and differ for each instrument combination (not given).

The Referee is correct in the fact that the radiosonde profiles smoothed using the four Akernels can be very different from each other. In the revised version of the manuscript we do note that *"while comparison of the retrievals to the relative Akernel-smoothed radiosonde profiles can be used to minimize the vertical representativeness effects due to the different vertical resolutions of these profiles, we note that a statistical comparison between the four configurations of the observational vector would not be fair if each of their retrieved profiles is compared to a different Akernel-smoothed radiosonde profile. Therefore, in the statistical analysis presented later in the manuscript (section 4.2), mean bias, root mean square error (RMSE), and Pearson correlation coefficients will be computed between the various MWR's retrieval configurations and the unsmoothed radiosonde profiles, just interpolated to the same vertical levels of the retrieved profiles."*

Lines 485-487: scientific reasoning missing

**Commented [4]:** Text on lines 485-487 was: "However, one radiosonde profile showed a large bias (> 5 oC) against all seven levels of BAO temperature measurements and against all PRs, therefore we decided to exclude this particular radiosonde profile from the statistical calculations."

We included the following text in the revised version of the manuscript: *"However, **one radiosonde profile showed a large bias (> 5 ºC) against all seven levels of BAO temperature measurements and all available Tv measurements from the RASS 915 (eight measurements up to 600 m AGL) and from the RASS 449 (nine measurements up to 1100 m AGL), therefore this particular radiosonde profile was excluded from the statistical analysis** (Figure is included). Moreover, while the RASS data could be collected properly under rainy conditions, MWR data could be potentially deteriorated due to water present on the radome, therefore six profiles (three for March 13, and one for May 1, 3 and 4) were eliminated from the statistical evaluation. These restrictions lowered the number of total radiosonde launches used to 52."*

[Figure]

Conclusions: Argumentations are in general not comprehensible

The Conclusion Section has now been rewritten and we do hope that the summary we present is now clearly understandable.

Lines 737-741: "Of the PRs configurations tested, we find better statistical agreement with the radiosonde observations when the RASS 449 is used together with the surface observations and brightness temperature from only the zenith MWR observations and doubling the random radiometric uncertainty on the MWR observations (MWRz2sigma449) relative to the uncertainty calculated over the selected clear-sky days." I cannot figure out what this sentence is supposed to say.

As mentioned above, the Conclusion Section has been completely rewritten, hopefully being clearly understandable now. Also, the text mentioned by the Referee is now deleted because we decided to completely remove the analysis presented before where the random radiometric uncertainty on the MWR observations relative to the uncertainty calculated over the selected clear-sky days was doubled (MWRz2sigma449). We decided to remove this part because we realized the motivation was too specific to this dataset, but wouldn't be useful in general, on another dataset.

Lines 744-746: "The larger assumed radiometric uncertainty in the MWR Tb observations allows the retrieval to overcome both (a) the small systematic errors that exist between the MWR observed Tb values and the RASS measurements and (b) the systematic errors that exist in forward microwave radiative models (Cimini et al. 2018)." How can there be "small systematic errors" between MWR TBs and RASS measurements? They are of different physical dimensions. What systematic errors exist in the forward radiative transfer model, how large are they and how do we know that the PR can overcome them?

As mentioned in the above answer, the text mentioned by the Referee is now deleted because we decided to completely remove the analysis presented before where the random radiometric uncertainty on the MWR observations relative to the uncertainty calculated over the selected clear-sky days was doubled (MWRz2sigma449).

---

## Author Response (AR3)

**Review AMT-2021-9:**

**Improving thermodynamic profile retrievals from microwave radiometers by including Radio Acoustic Sounding System (RASS) Observations**

**ANSWERS to REFEREE 1**

We wish to thank very much both Referees for carefully re-reading our manuscript for the third time and for offering additional comments towards its improvement. We believe all of your notes and recommendations are very useful. We hope that our latest changes will make our manuscript clearer and more transparent. Please, find below our point-to-point answers in red.

**Minor corrections:**

**Line 337**: we have demonstrated the augmentation → we have demonstrated the extension. Done.

**Line 447**: It is mentionned that the BC is computed with the TROPOe retrievals. I doubt this would be a good idea as the TROPOe retrieval is obtained to minimize the distance with the observation. So, if the observations is biased, we of course end with an atmospheric profile compensating the bias in the observation. Thus, I assume that the difference between the observation and the simulation from the TROPOe retrievals should tend to zero as the PR want to minimize this distance. I think the authors wanted to mention TOPROe background profiles from the climatology as it is confirmed in line 498 and explained in their reviewer's answers. Please correct the text accordingly.

We have found that there are often spectral features in the observed minus computed brightness temperature residuals that could not be explained by any physically realistic atmospheric profiles, and can only result because of a calibration error in the observations. This TROPoe bias-correction method is aimed purely to remove this unphysical spectral signature. We realize that this bias-correction approach could introduce a bias in the retrieved temperature and humidity profiles. The more appropriate method to determine this spectral bias correction is with using independent radiosondes; however, these are not always available which is why we wanted to present both bias correction methods in this paper.

The text in the paper is changed (additional text is highlighted):

"While this radiosonde BC method can be employed for the XPIA dataset, for other campaigns this approach would not be possible if co-located radiosonde observations were not available. For this situation, an alternative method for correcting the MWR Tb biases is presented. **There are often spectral features in the observed minus computed brightness temperature residuals that could not be explained by any physically realistic atmospheric profiles, and can only result because of a calibration**

**error in the observations. This alternative bias-correction method is aimed purely to remove this unphysical spectral signature.**"

"The Tb bias is then computed for each of the 22 channels as the averaged difference between the observed Tb from the MWR zenith observations and the forward model calculated Tbs at zenith using the TROPoe-retrieved profiles (Y1) of those selected clear-sky days. This method identified spectral calibration errors in the MWR observations that could not be explained by physically realistic atmospheric profiles. This bias-correction technique, **which accounts for those unphysical spectral calibration features,** will be referred to as 'TROPoe BC'."

**Line 635**: as a function of the height → as a function of height

We found two places in the paper with this phrase (but not on line 635) and changed them accordingly.

**Figure 8**: bottom right: change 0-5 km averaged into 0 – 3 km averaged.

Sorry, we cannot find this text in the "Retrievals_paper_review2_final.pdf (or docx)", it was in the previous version.

**Review AMT-2021-9:**

**Improving thermodynamic profile retrievals from microwave radiometers by including Radio Acoustic Sounding System (RASS) Observations**

**ANSWERS to REFEREE 2**

We wish to thank very much both Referees for carefully re-reading our manuscript for the third time and for offering additional comments towards its improvement. We believe all of your notes and recommendations are very useful. We hope that our latest changes will make our manuscript clearer and more transparent. Please, find below our point-to-point answers in red.

- Line 105: "atmospheric temperature and humidity content" -> "atmospheric temperature, humidity, and liquid water content"

MWRs are sensitive to the LWP, not the LWC, therefore we use "atmospheric temperature, humidity, and liquid water path (LWP)".

- Lines 137-140: I suggest replacing the following sentence:
"is still limited, being a function of both radar frequency and atmospheric conditions (May and Wilczak, 1993). It is determined both by the attenuation of the sound, which is a function of atmospheric temperature, humidity, and frequency of the sound source,

and the advection of the propagating sound wave out of the radar's field-of-view"
with:
"is limited by sound attenuation, which is a function of both radar frequency and atmospheric conditions (May and Wilczak, 1993) such as temperature, humidity, and the advection of the propagating sound wave out of the radar's field-of-view"

Done.

- Lines 146: "has been" -> "has been and still is"

Done.

- Line 244: "inline" - not sure this is the proper word.

The word "inline" is deleted.

- Lines 303-307: I'm missing the introduction of the forward model for Tv and its Jacobian (at lines 361-365).

MonoRTM is used as the forward model only for Tb, Tv in TROPoe is directly computed from the ambient temperature and water vapor mixing ratio (using basic thermodynamic equations). Similarly, the Jacobian is computed analytically from this equation.

- Lines 320-323: I'm missing how sigma_Tb and sigma_Tv are estimated to be included in Se.

Sigma_Tb (lines 355-359) is set to the standard deviation from a detrended time-series analysis for each channel during clear-sky time frame, smaller for K-band channels and larger for V-band channels. Sigma_Tv (lines 362-364) is calculated based on SNR, smaller for high SNR and larger for low SNR.

- Lines 328-333: I'm missing how LWP is estimated to be included in Sa.

We make an assumption for LWP, but the uncertainty in the assumed LWP value in the prior is very large as to have no impact (constraint) on the retrieval. On page 16, lines 330-331, we added the sentence:

"LWP is arbitrarily assigned in Xa, with large values chosen for its uncertainty in Sa, so that it does not impact (constrain) the retrieval"

- Line 382: I suggest replacing "to reduce the degrees of freedom associated with clouds" with "to eliminate uncertainties associated to clouds"

Done.

- Line 404: Maybe the authors mean the TOPROe background profiles from climatology? This is also noted by the reviewer. Please modify the text accordingly.

We already explained to referee 1 why we included the TROPoe bias-correction technique in this paper. We understand that using independent radiosondes is more appropriate for spectral bias correction. However, radiosonde data are not always available. Additionally, there are often spectral features in the observed minus computed brightness temperature residuals that could not be explained by any physically realistic atmospheric profiles. TROPoe bias-correction method is aimed purely to remove this unphysical spectral signature.

The text in the paper is changed (additional text is highlighted):

"While this radiosonde BC method can be employed for the XPIA dataset, for other campaigns this approach would not be possible if co-located radiosonde observations were not available. For this situation, an alternative method for correcting the MWR Tb biases is presented. **There are often spectral features in the observed minus computed brightness temperature residuals that could not be explained by any physically realistic atmospheric profiles, and can only result because of a calibration error in the observations. This alternative bias-correction method is aimed purely to remove this unphysical spectral signature.**"

"The Tb bias is then computed for each of the 22 channels as the averaged difference between the observed Tb from the MWR zenith observations and the forward model calculated Tbs at zenith using the TROPoe-retrieved profiles (Y1) of those selected clear-sky days. This method identified spectral calibration errors in the MWR observations that could not be explained by physically realistic atmospheric profiles. This bias-correction technique**, which accounts for those unphysical spectral calibration features,** will be referred to as 'TROPoe BC'."

- Lines 714-720 (and in general): References to figure formatting (panel position, line color, marker type, etc) could be removed from main text and left in the figure caption only.

This text is changed.

- Line 736: Any comment on the why RASS 449 Tv bias seems to depart above 1.3 km?

Data availability is getting smaller with height, but the Tv bias is around 0.2 C at that height and should be considered small.

- Lines 741-742: I suggest adding the sentence:
"the RASS biases, because of the combined information from RASS and MWR."

Done.

- Line 836: "and above around 1.5 km AGL" -> "and only between 1.5 and 3 km AGL"

Done.

---

## Author Response (AR4)

**Review AMT-2021-9:**

**Improving thermodynamic profile retrievals from microwave radiometers by including Radio Acoustic Sounding System (RASS) Observations**

**ANSWERS to REFEREE**

We wish to thank you very much for the additional re-reading our manuscript and for offering two comments towards its improvement. We believe these notes are useful. We hope that these changes will make our manuscript clearer and more transparent. Please, find below our answers in red.

**Minor corrections:**

**Line 330**:

- Please, make explicit the number used for LWP uncertainty into the following sentence, as "large" is only relative: "LWP is arbitrarily assigned in Xa, with large values chosen for its uncertainty in Sa, so that it does not impact (constrain) the retrieval"

We included additional sentence: "**Presently, the assumed uncertainty in LWP in the prior is assigned to 200 g/m^2 in TROPoe configuration file**".

**Fig. 1A:**

- Please, add the retrieval performances from non-BC MWRzo/MWRz to Figure 1A. As is, the manuscript gives no evidence that TROPoe BC provides better retrievals than original non-BC Tb. Actually, Fig.1A suggests that TROPoe BC provides no better retrievals than NN (based on non-BC Tb), specially in the boundary layer. If the TROPoe BC does not improve performances significantly with respect to non-BC MWRzo/MWRz, that should be clearly stated. In such a case, I'd suggest mentioning alternative BC sources to be considered, e.g. based on NWP profiles (as used in https://doi.org/10.5194/amt-13-6593-2020 and references therein).

We made two additional sets of TROPoe retrievals for zenith and zenith-oblique scans without any bias-correction and included those two new statistics profiles in Fig. 1A (below) and included the text:

"**The PRs without any Tb bias-correction (dashed lines in Fig. 1A) clearly indicate that the BC is useful and needed, showing very noticeable degradation in all three statistical measures above 3 km, and larger RMSE and bias in 0.5-1.5 km AGL compared to TROPoe BC method**".

**Temperature (C)**

[Figure]

*Fig. 1A. Pearson correlation, RMSE, and mean bias for temperature profiles for MWRz in grey (and purple) and MWRzo in black (and maroon) when the radiosonde BC (and the TROPoe BC) method is applied. **TROPoe temperature retrievals without any bias-correction are shown for MWRz in dashed purple and for MWRzo in dashed maroon.** Included in this figure are the NN temperature profiles, from the zenith scan (in beige), and from the averaged oblique scans (in green).*